# Dynamic changes on Wilkins Ice Shelf during the 2006-2009 retreat derived from satellite observations

Melanie Rankl[1], Johannes Jakob Fürst[1], Angelika Humbert[2/3], Matthias Holger Braun[1]

[1]Friedrich-Alexander Universität Erlangen-Nürnberg, Institute of Geography, 91058 Erlangen, Germany
[2]Alfred Wegener Institute, Helmholtz Centre for Polar and Marine Research, Glaciology Section, 27568 Bremerhaven, Germany
[3]University of Bremen, Department of Geosciences, 28359 Bremen, Germany

*Correspondence to*: Melanie Rankl (melanie.rankl@fau.de), Johannes Fürst (johannes.fuerst@fau.de)

**Abstract**. The vast ice shelves around Antarctica provide significant restraint to the outflow from adjacent tributary glaciers. This important buttressing effect became apparent in the last decades, when outlet glaciers accelerated considerably after several ice shelves were lost on the Antarctic Peninsula. The present study aims to assess dynamic changes on Wilkins Ice Shelf during a series of ice-front retreat and partial collapse between early 2008 and 2009. The total ice-shelf area lost in these events was 2135 ± 75 km² (~15% of the ice-shelf area relative to 2007). Here, we use time-series of Synthetic Aperture Radar (SAR) satellite observations (1994/96, 2006-2010) in order to derive variations in surface-flow speed from intensity-offset tracking. Spatial patterns of horizontal strain-rate, stress and stress-flow angle distributions are determined during different ice-front retreat stages. Prior to the final break-up of an ice bridge in 2008, a strong speed-up is observed, which is also discernible from other derived quantities. We identify areas that are important for buttressing and areas prone to fracturing using in-flow and first principal strain-rates as well as principal stress components. Further propagation of fractures can be explained as the first principal components of strain-rates and stresses exceed documented threshold values. Positive second principal stresses are another scale-free indicator for ice-shelf areas, where fractures preferentially open. Second principal strain-rates are found to be insensitive to ice-front retreat or fracturing. Changes in stress-flow angles highlight similar areas as the in-flow strain-rates, but are difficult to interpret. Our study reveals the large potential of modern SAR satellite time-series to better understand dynamic and structural changes during ice-shelf retreat, but also points to uncertainties introduced by the methods applied.

## 1 Introduction

During the last few decades several vast floating ice shelves have been lost (Cook and Vaughan, 2010; Doake and Vaughan, 1991; Rott et al., 2002; Scambos et al., 2003, 2000), which entailed significant speed-up and thinning of previously restrained outlet glaciers (Berthier et al., 2012; Rignot, 2004; Scambos, 2004). The reason is that ice shelves restrain the outflow from tributary glaciers (Dupont et al., 2005), thus they serve as natural buttresses (De Angelis and Skvarca, 2003; Khazendar et al., 2015; Rott et al., 2002; Scambos et al., 2014). On the Antarctic Peninsula (AP), 12 major ice shelves have either disintegrated or significantly retreated (Cook and Vaughan, 2010; Doake and Vaughan, 1991; Rott et al., 2002; Scambos et al., 2003, 2000). In addition, many of these ice shelves experienced important thinning over the past two decades (Fricker and Padman, 2012; Paolo et al., 2015; Wouters et al., 2015; Zwally et al., 2005). Ice shelves along the western coast of the AP exhibit thinning rates (Paolo et al., 2015) that are often twice as high as those on the eastern side (e.g., Larsen Ice Shelf). Over 85% of the tributary glaciers showed retreating tongues (Cook, 2005). Reasons for this regionally concentrated ice shelf recession are manifold. Explanations range from a warming atmosphere (O'Donnell et al., 2011; Steig et al., 2009; Vaughan et al., 2003), to a reduction in sea-ice coverage on the western side of the AP (Stammerjohn et al., 2008), to rising ocean temperatures on the Bellingshausen Sea continental shelf (Martinson et al., 2008; Meredith and King, 2005) and to warming deep waters in the Weddell Sea (Robertson et al., 2002).

There were regular attempts to assess ice shelves' structural stability under ice-front recession. A first criterion was formulated after the final collapse of Larsen A and during the subsequent gradual recession of Larsen B. Doake et al. (1998) inferred the two principal strain-rate components in horizontal direction. The smaller, second principal strain-rate separates areas of purely extensive ice flow from more constrained regions. A 'compressive arch' was forwarded to delineate a threshold line, beyond which
ice shelves are expected to ultimately collapse. In the meantime, ice-flow models were successfully applied to get a more comprehensive description, including the stress distribution. Studying Larsen C ice shelf, Kulessa et al. (2014) proposed that ice-shelf stability should be assessed from the stress-flow angles, i.e., the angle between the ice-flow direction and the first principal stress. When the two fields align and the angle tends to zero, fracture-opening rates were considered to maximize. If angles are small near an ice front, the ice-shelf geometry would be considered unstable.

Also relying on the stress regime within the ice, Fürst et al. (2016) quantified the maximum buttressing potential of all ice shelves in Antarctica. Though not addressing any stability criteria, they could show that ~13% of all floating ice is dynamically not relevant. Once ice-front recession exceeds this passive ice-shelf area, dynamic consequences are expected.

The aim of this study is to assess changes in the dynamic state of Wilkins Ice Shelf (WIS) during the multi-stage break-up over the last decade. For this purpose, ice-velocity maps are inferred from radar satellite observations in 1994/96 and from 2006-2010.
From these maps, we derive time-series of strain-rates, the principal stress distribution and of stress-flow angles. Changes in these time-series are discussed during the different collapse stages on WIS, which allows a first assessment of previously invoked stability criteria (Doake et al., 1998; Fürst et al., 2016; Kulessa et al., 2014).

## 2 Regional settings and retreat history of Wilkins Ice Shelf

WIS is located at the south-western part of the AP covering an area of 10,150 km² in April 2015 (Fig. 1a). The ice shelf is confined
by Alexander, Rothschild and Latady Island. Mass gain at WIS is dominated by surface accumulation (Vaughan et al., 1993). Further contribution of mass originates from the main in-flow of Lewis Snowfield south-west of the ice shelf. Contribution of mass from Gilbert Glacier and two inlets, called Schubert and Haydn inlets, is highly restrained by prominent ice rises (e.g. Dorsey Island).

During the last decades, WIS has undergone significant dynamic changes: Surface lowering of in total -4.95 ± 0.21 m between
1978 and 2008 on the southern WIS was found to be the largest for all AP ice shelves (Fricker and Padman, 2012). In recent years (2000-2008), however, thinning rates were less negative or even slightly positive on the northern WIS (Paolo et al., 2015). Average thickness changes of WIS were -0.62 ± 0.12 m a$^{-1}$ for the period 1994-2012 derived from radar altimetry and were attributed to basal melting (Paolo et al., 2015). Basal melt rates of WIS were estimated by several authors: Holland et al. (2010) found 0.66 m a$^{-1}$ averaged over the period 1979 to 2007, Padman et al. (2012) inferred rates of 1.3 ± 0.4 m a$^{-1}$ during 1992-2008, while Rignot
et al. (2013) derived 2003-2008 average rates of 1.5 ± 1 m a$^{-1}$.

In addition, the WIS underwent major ice-front retreat (Fig. 1a; Braun et al., 2009; Scambos et al., 2009, 2000). Based on the analysis of time-series of satellite imagery and historic maps, numerous break-up stages were quantified (Arigony-Neto et al., 2014). Several causes for break-up at WIS were proposed in the literature: fracture formation due to bending stresses in combination with surface melt or brine infiltration (Scambos et al., 2009), transoceanic infragravity waves which induce fractures
at Antarctic ice shelves (Bromirski et al., 2010) and break-up due to bending stresses caused by enhanced basal melt (Braun and Humbert, 2009; Humbert et al., 2010). The ice-front retreat led to the formation of a remnant ice bridge between Latady and Charcot Island on the western WIS (ice bridge in Fig. 1a), which collapsed in April 2009. Braun et al. (2009) also documented the development of fractures and rifts as well as the role of ice rises in the break-up processes on WIS. Between 28/29 February and

30/31 May 2008, an area of 585 km² broke-off narrowing the connection between these two confining islands (Braun et al., 2009). In June/July 2008, an area of $1220 \pm 75$ km² (~8% of the ice-shelf area in respect to 2007) was lost at the eastern side of this ice bridge (Fig. 1a), implying further weakening (Humbert and Braun, 2008). Braun and Humbert (2009) found highly variable ice thicknesses (~250-170 m) along the ice bridge implying important buoyancy differences. The break-up events in May and June/July

2008 have demonstrated that such events can occur in austral winter (Braun et al., 2009). This contradicts previous assumptions that ice-shelf break-up mainly depends on summer-surface melt ponds (Doake and Vaughan, 1991; MacAyeal et al., 2003; Scambos et al., 2000). In early April 2009, the final ice-bridge collapse corresponded to an area loss of 330 km². More than 100 tabular icebergs calved off. After the collapse of the ice bridge, small-size calving events occurred, which primarily took place along the south-west ice front between Latady Island and Lewis Snowfield.

**3 Material and Methods**

**3.1 Surface velocities**

Surface velocities of the ice shelf and its tributary glaciers were derived from SAR (Synthetic Aperture Radar) intensity-offset tracking (Strozzi et al., 2002) using repeat ALOS PALSAR (46 day time interval) Single Look Complex (SLC) image pairs (Table S1). Results in March 1994/96 were taken from InSAR (Interferometric Synthetic Aperture Radar) derived surface flow presented

in Braun et al. (2009) and from ERS-1/-2 (35 day time interval) intensity-offset tracking. The latter technique cross-correlates the backscatter intensity pattern of a pair of SAR images of different acquisitions dates. For this purpose, small image patches are shifted over the entire image (Table 1) and for each patch, the maximum of the 2-D cross-correlation function yields the image offsets in range and azimuth directions. If coherence between both image patches is retained, the speckle pattern is additionally correlated. Offsets of minor confidence were rejected based on a signal-to-noise-ratio (SNR $\leq 4$). The processing was performed

using Gamma Remote Sensing software (Werner et al., 2000). Geocoding of the final range and azimuth offsets from SAR to map geometry was based on the WGS84 ellipsoid. The spatial gridding was set to 50x50 m, which is also true for the derived quantities in 3.2.

The method relies on surface patterns, which are identifiable in both images. However, co-registration and intensity-offset tracking performed on single scenes of the nearly structure-less ice shelf was rarely successful. Therefore, single scenes were concatenated

along-track (Fig. S1). Additionally, we used a binary mask of very slow/non-moving (e.g., ice rises, bedrock) and moving areas (ice shelf, tributary glaciers, sea) to perform co-registration on stable areas only (Fig. S1). In a post-processing step, the flow magnitude and direction were filtered using the approach described in Burgess et al. (2012). By using a 5 x 5 pixel moving window approach, displacement vectors were discarded iteratively when deviating more than 30% from the median length of the window's centre vector or when deviating from a predefined orientation of the centre vector (thresholds 20°, 18° and 12°).

For each year, several displacement fields were mosaicked. The mosaicked surface flow shows slight deviations in the flow magnitude along the boundaries of each satellite flight path. These offsets might be due to short-term variations of ice flow between image acquisitions, processing artefacts or due to varying co-registration accuracies related to the restricted availability of non-moving areas in each scene. The magnitude of these offsets is non-linear and ranges between ~3 and 18 m yr$^{-1}$ on the main ice-shelf area. The offsets are larger close to the ice front, where the displacement fields capture the short-term motion of the ice

mélange. The derived flow fields were not corrected for these non-linear offsets, however, the estimated co-registration accuracy in Table 2 accounts for these deviations (see below). Flow differences in 2007/2008 and 2008/2009 were calculated for the south-western part of the ice shelf only, since this part reveals the most significant dynamic changes.

The estimation of errors in the derived velocity fields was done as described in McNabb et al. (2012) and in Seehaus et al. (2015). For each velocity field a value based on the accuracy of the co-registration ($\sigma_v{}^C$) was calculated and a second value ($\sigma_v{}^T$) described uncertainties involved in the intensity-offset tracking algorithm (Table 2). Further error contribution related to the orbital information of the image acquisitions or the atmospheric influence are still difficult to quantify. The magnitude of the term $\sigma_v{}^C$ was derived from the median of the velocities over non-moving areas (based on up to 25,000 samples per image pair), e.g., ice rises or bedrock, where zero ice motion is assumed. The error estimation over non-moving ground is a standard procedures when using intensity-offset tracking for ice velocity determination (e.g., Burgess et al., 2012; McNabb et al., 2012; Quincey et al., 2009, 2011; Seehaus et al., 2015). Since no additional calibration of the derived offset fields over stable ground has been undertaken, the term $\sigma_v{}^C$ captures all errors related to the co-registration procedure. The second term $\sigma_v{}^T$ describes uncertainties related to the intensity-tracking algorithm, the spatial resolution and the time interval between image acquisitions. It is calculated using

$$\sigma_v^T = \frac{C \Delta x}{z \Delta t} \qquad \text{(Seehaus et al., 2015).} \qquad (1)$$

$C$ describes the uncertainty of the tracking algorithm ($C$=0.4), $\Delta x$ the image resolution in ground range, $z$ the oversampling factor used in the tracking process and $\Delta t$ the time period between image acquisitions. The final error estimate $\sigma_v$ is derived from the sum of both terms $\sigma_v{}^C$ and $\sigma_v{}^T$ (Table 2).

When calculating the strain-rate and stress components from the displacement fields (see 3.2), a wavelike pattern emerges in 2006, 2008 and 2009, which dominates in areas where flow speeds are small. This pattern was detected in comparable studies calculating surface velocities from intensity-offset tracking (Joughin, 2002; Nagler et al., 2015). It was attributed to fluctuations in the polar ionospheric electron density and may affect the phase measurement of a SAR sensor, but also the correct mapping of the azimuth pixels' position (Gray et al., 2000). This effect is found to be larger for L-band than for C-band acquisitions. The wavelength of this pattern in L-band frequencies was scaled to 5-10 km (Gray et al., 2000), which is comparable to the pattern visible in Figures 3 and 4. A smoothing of this pattern by averaging several flow fields over multiple acquisitions as proposed in Nagler et al. (2015) is impossible on WIS due to lack of further, suitable image pairs. Another study found variations in the tropospheric water content influencing the measured path delay with InSAR (Drews et al., 2009). However, this effect was restricted to C-band InSAR (Williams et al., 1998) and no influence on the image intensity is known. Hence, ionospheric disturbances remain a likely explanation for the detected wavelike pattern in this study. However, as the pattern has some link to the structure of the ice shelf and is persistent over years, we cannot rule out completely that it is a real feature of the displacement field.

## 3.2 Stress tensor and strain-rates

In order to infer the components of the stress tensor associated with a flow field, a constitutive equation is required. In glaciology, ice is typically described as a non-Newtonian fluid with a viscosity $\eta$ depending on temperature and effective strain-rate. The constitutive equation for the deviatoric stress $\tau_{ij}$ can be written as follows:

$$\tau_{ij} = 2\eta \cdot \dot{\varepsilon}_{ij} \qquad (2)$$

Here, $\dot{\boldsymbol{\varepsilon}}$ is the strain-rate tensor, which holds information on spatial derivatives of the horizontal velocity field:

$$\dot{\varepsilon}_{ij} = \frac{1}{2}\left(\frac{\partial u_i}{\partial x_j} + \frac{\partial u_j}{\partial x_i}\right) \qquad (3)$$

The strain-rate tensor describes how much the ice is deformed at a certain location and its horizontal components can be inferred from surface velocities. For the purpose of understanding the changes in the stress regime due to the break-up events, it is of particular importance to look at strain-rates and not velocity fields. Velocity fields are derived by measuring displacements of features assuming that the features themselves do not change. In case of opening of a rift, the feature is often a rift face or a crack front, and both change over time. Thus, the interpretation of the displacement field as a flow velocity can only be done in areas

without rift opening. Otherwise, the displacement due to opening of a rift and the creep of ice are superimposed in the velocity field. Strain-rates, however, overcome this flaw, as they are spatial derivatives.

The ice viscosity determines how strain-rates translate into deviatoric stresses $\tau_{ij}$. Stress deviators are linked to the full Cauchy stress $\boldsymbol{\sigma}$ via the isometric pressure.

$$\tau_{ij} = \sigma_{ij} - \frac{1}{3}\delta_{ij}\sum_k \sigma_{kk} \qquad \text{(Cuffey and Paterson, 2010)} \tag{4}$$

At the ice-shelf surface (denoted as superscript $s$ in the following), where satellite data were acquired, the relation between deviatoric and full stresses simplifies to:

$$\sigma_{xx}^s = 2\tau_{xx}^s + \tau_{yy}^s \tag{5}$$

$$\sigma_{yy}^s = \tau_{xx}^s + 2\tau_{yy}^s \tag{6}$$

This assumes that bridging effects are negligible near the ice surface. Given the fact that we observe the horizontal velocities on the ice-shelf surface, we can compute from (5) and (6) the stress components on the surface. In the following, however, we will omit specifying this limitation and simply speak of strain and stress conditions.

The viscous response of an ice body is non-linear and depends itself on the strain regime:

$$\eta = \frac{1}{2}mA^{-1/n}\cdot\dot{\varepsilon}_e^{(1-n)/n} \tag{7}$$

Here, $\dot{\varepsilon}_e^2 = \frac{1}{2}\left[tr(\dot{\varepsilon}^2) - (tr(\dot{\varepsilon}))^2\right]$ is the second invariant of the strain-rate tensor, with trace defined as: $tr(\dot{\varepsilon}) = \sum_{i=1}^{3}\dot{\varepsilon}_{ii}$. We assume isotropic material properties and a flow exponent of $n = 3$. $A$ is the rate factor, which determines the readiness of the viscous material to deform under a given stress (Van der Veen, 2013). For WIS, we assume a constant rate factor $A$ of 1.7 x 10$^{-24}$ Pa$^{-3}$ s$^{-1}$ (Cuffey and Paterson, 2010) corresponding to near-temperate ice of about -2° C as observed in this area (Braun et al., 2009). The rate factor is a function of ice temperature and microscopic water content and consequently can show some spatial variability. The

enhancement factor $m$ is assumed constant and set to 1. In general, $m$ is, however, a function of grain size, impurities, damage degree and other variables (Cuffey and Paterson, 2010, p. 71). These additional factors are difficult to account for and are therefore not reflected in the subsequently presented stress fields.

Strain and stress components are real and respective tensors are symmetric which ultimately reflects the conservation of angular momentum in continuum mechanics. As a consequence, all eigenvalues are real. First and second principal components span the

range of minimal and maximal extensive or compressive strain-rates or stress occurring on the ice surface (Gross et al., 2009). The first principal strain-rate gives maximal extension rates and is therefore of interest to define a crevasse initiation or material failure criteria (Benn et al., 2007; Vaughan, 1993).

The strain and stresses components in this study will be used in two regards: (i) for assessing temporal changes in them after break-up events and (ii) to assess which area of the ice shelf might reach a critical limit and be prone to new crack initiation. Suggested

threshold strain-rates for fracture initiation span a wide range from 0.01 a$^{-1}$ (2.7·10$^{-5}$ day$^{-1}$) on Greenland (Meier, 1958) to 0.004 a$^{-1}$ (1.0·10$^{-5}$ day$^{-1}$) on White Glacier, Canada (Hambrey and Müller, 1978). To describe damage initiation within an ice body, Krug et al. (2014) used a threshold criterion for the first principal stress component. In terms of stresses, inferred thresholds range from 90 to 320 kPa (Vaughan, 1993). The Hayhurst rupture criterion defines a stress threshold of 330 kPa for fracture initiation (Pralong et al., 2006). Second principal surface stress, however, was recognized to be important in terms of ice-shelf stability (Doake et al.,

1998) or, more general, to assess and quantify the dynamic susceptibility of ice shelves (Fürst et al., 2016).

## 4 Results

In the following, dynamic consequences of the successive retreat stages of WIS are analysed on the basis of multi-temporal maps of surface velocities, strain-rates, principal surface stresses and the stress-flow angle criterion (Figs. 2-4). First, we will give a short overview of the figures stating the importance and caveats of the presented velocity data and the derived quantities (Sect. 4.1).

Then, we will use all fields to analyse the dynamic re-organisation of the ice shelf during the different retreat stages (Sect. 4.2). Finally, we will investigate which quantities are most suitable for identifying ice-shelf instability (Sect. 4.3).

### 4.1 Introduction to analysed quantities

We present a unique sequence of multi-temporal surface-velocity maps for WIS spanning the period 1994/96 to 2010 (Figs. 2 & 3). During all years, inflows from Gilbert Glacier, Haydn and Schubert inlets as well as originating from Lewis Snowfield are

clearly visible. For the grounded part of Gilbert Glacier, surface velocities readily exceed 300 m yr$^{-1}$. Yet, seawards of the grounding line, velocity magnitudes decrease abruptly. Inflow from Gilbert Glacier and Haydn inlet is prominently restrained by Dorsey Island. Further west, additional restraint is provided by the Petrie and Vere ice rises as well as by Latady Island. This initial deceleration is characteristic for all major outlet glaciers draining into WIS and illustrates that ice-shelf flow is highly constrained by the geometric setting (islands, ice rises and pinning points). Dependent on the consecutive image pair (Table 2), the velocity

error associated to the intensity-offset tracking method falls into a range between 25 to 100 m yr$^{-1}$. In 1994/96, errors and velocity magnitudes are comparable on the ice shelf (Fig. 2a). In more recent years, velocities are found to be higher and their magnitudes are thus, relatively more reliable. The 2006/2007 acceleration shows a meaningful pattern with largest values near the northern-western ice front. The relatively high error value associated with the velocity fields has, however, to be considered when interpreting all derived quantities as these rely on sensitive spatial velocity differences. Despite the velocity error, offset tracking

produces velocity fields representative for time periods covering many days or even months (Table 2). When we interpret and discuss these fields, it is essential to keep in mind that values might represent an average displacement and that we cannot discern between sudden or gradual flow re-organisation.

The first derived quantities are the strain-rates (Fig. 3). Positive values indicate an extensive flow regime, while negative values imply compressive flow. The reliability of these strain-rates is highly dependent on the quality of the inferred flow magnitude and

direction. Relative errors become dominant in areas where flow speeds are small. To avoid misinterpretation in such areas, we limit the following discussion to areas showing considerable motion ($\geq$ 0.05 m day$^{-1}$ or 20 m yr$^{-1}$), unless specified otherwise. Additionally, we calculated values at each point in a local coordinate system with the two axis aligned parallel and perpendicular to the local flow direction. In this way, we can determine the strain-rate in flow direction. Where ice flow is restrained by obstacles, such as ice rises, island and other pinning points, this strain-rate component is negative and thus, compressive in all years. (Fig.

3b, f, j, n). Additional inflow originating from Latady Island leads to compressive strain-rates further south (Fig. 3b, f, j, n). All strain-rate components are elevated, where fractures have formed or continue to open.

Based on the derived strain-rates and by assuming a constant rate factor, principal surface-stress components were computed (Fig. 4). The first principal stress gives the highest extensive stress acting within the material. Between 2006 and 2009, first principal stresses are generally positive (Fig. 4a, d, g, j). Values are often elevated nearby fractures. Moreover, local maxima are often

located at fracture tips, where extensive forces concentrate. Away from the ice bridge, the second principal stress component is mostly negative. This illustrates that ice flow is largely constrained by the geometric setting. As the magnitude of the stress field scales with the rate-factor choice, the latter discussion mostly focusses on the pattern and sign changes in the second principal stress field. Both are robust even if a different rate factor was chosen. Sign changes would even persist for a spatially variable rate factor field.

Finally, the angle between the flow direction and the first principal stress direction is computed following Kulessa et al. (2014). On WIS, we confirm that stress-flow angles are indeed small in areas where fractures formed. The derivation of stress-flow angles shown in Kulessa et al. (2014) is based on modelled ice-shelf velocities, flow lines and stresses using a continuum-mechanical ice-flow model. Here, however, the angle is calculated from observations, which results in a higher spatial variability of the underlying quantities.

## 4.2 Retreat stages

In March 1994/96, we find a stagnant ice shelf with surface velocities below 0.2 m day$^{-1}$ (70 m yr$^{-1}$) in most areas (Fig. 2a). Increased flow rates are visible downstream of Lewis Snowfield and Gilbert Glacier. Velocities increase up to 1 m day$^{-1}$ along the northern ice front near Charcot Island, which is likely associated with the 1993 break-up event there (remnant icebergs are still visible in the backscatter SAR image of Fig. 2a). Even though the northern ice front retreated significantly until 2006, no significant speed-up is observed (Fig. 2b). All along the elongated ice bridge connecting to Charcot Island, two distinct flow branches are discernible by a small step in velocity magnitude. The western branch shows higher velocities, which indicates that it already experiences and transmits less buttressing from Charcot Island. The disconnection between these two flow branches has likely started between February and June 1998, when a double fracture (~45 km long, green line in Fig. 3a) formed parallel to the western ice front (Braun and Humbert, 2009). The fracture is imprinted in all derived quantities and is most pronounced in the in-flow and first principal strain-rate components as well as in the first principal stress component (Fig. 3b, c and 4a). For the second principal stress component, the fracture is highlighted by a sign change to positive values (a fully tensile stress regime), while most of the remaining ice shelf shows a partially compressive regime.

In July 2007, the western branch of the ice bridge further accelerated in response to the formation of a second, longitudinal fracture pair (~37 km in length, purple line in Fig. 3e, Braun et al., 2009). A clear distinction of the acceleration in two branches is possible, with more elevated values along the western side of the bridge (Fig. 3e). Most of the derived quantities change in reaction to the opening of the fracture pair. Stress and strain-rate components show increased values and often pronounced maxima, while the stress-flow angle tends to smaller values (Figs. 3f, g & 4d, e, f). From the in-flow strain-rates and the second principal stress we observe that a narrow region on the eastern branch shows very negative values indicating important compression. Away from the ice bridge, all fields remain similar to the 2006 situation, which implies that there is no clear indication for an imminent dynamic re-arrangement. Buttressing due to e.g., ice rises is well expressed in low values in principal stresses as well as in the in-flow and second principal strain-rates (Figs 3f, h & 4d, e).

Between February and July 2008, the western branch of the ice bridge broke-off and an area of 1805 ± 75 km² (~12% of the former ice-shelf area) was lost (Braun and Humbert, 2009; Humbert et al., 2010; Scambos et al., 2009). The consecutive velocity increase was highest on the ice bridge (velocities nearly tripled at the narrowest part) and reached as far upstream as the Petrie Ice Rises (Fig. 2g). From the stress and strain fields in 2007, it seems unlikely that this acceleration was triggered by the collapse of the western bridge, because no significant restraint to the ice flow was transmitted via this branch. It seems more likely that the eastern branch weakened significantly by the loss of a small, but important area near the 1998 fracture pair. This area provided significant buttressing in 2007 (e.g., Fig. 3f). In 2008, the flow regime has significantly changed showing a steep increase in velocities along several line segments perpendicular to the ice-bridge orientation (passing by Vere Ice Rise). These discontinuities are ultimately transmitted to all derived quantities but are least expressed in the second principal strain component (Fig. 3l). In the homologue stress component, values turn positive in this area as compared to the 2007 state (Fig. 4g, h). Turning towards the remaining ice bridge, the in-flow strain-rates show different signs on each side of the narrowest bridge segment (Fig 3j). These observations explain why an initial crack formed there and the later failure position of the bridge (Humbert et al., 2010). The prevailing

atmospheric circulation pushed the brittle ice mélange westward, which was the reason why the bridge yielded in this direction (Humbert et al., 2010).

The narrow remainder of the ice bridge finally collapsed in April 2009 and an area of 330 km² broke-off (Humbert et al., 2010). This final event caused no further acceleration on the remnant WIS (Fig. 2h). This suggests that the lost ice-shelf portion did not transmit or provide much buttressing to the central ice-shelf unit. No field gives indication for further fracture opening.

In 2010, results from intensity-offset tracking were unavailable for some parts of the WIS and the coverage was incomplete (Fig. 2f). Yet, from the covered ice-shelf area, we are confident that ice flow did not change significantly between 2009 and 2010 (Fig. 2e, f). In addition, the ice-front position remained stable during this period (Fig. 2f).

The future stability of WIS is considered rather weak. Based on the analysis of several SAR images acquired in recent years (2011-2015) the formation of fractures at the south-western ice front was detected (Fig. 1b). These fractures have developed since 2011 and have grown perpendicular to the south-western ice front (Fig. 1b). In August 2014, one of these fractures grew further towards the centre of the ice shelf, forming a kink parallel to the south-western ice front. Another fracture was detected on a Sentinel-1A imagery from April 2015, north of the ones described before (Fig. 1b). This fracture emerged perpendicular to the ice front towards the ice-shelf centre. Further growth of the fractures towards the ice-shelf centre and a connection of them, might lead to future loss of a large portion of the ice shelf and presumably a disconnection from Latady Island.

### 4.3 A-priori indications for retreat and fracturing

In this section, we want to investigate, which fields are most informative, when it comes to identifying weaknesses and predicting the next break-up stage. In 2006, no field shows any indication that another fracture pair would open. Yet, the existing fracture pair that formed in 1998 is discernible in most fields. This might simply imply that the integrity of the ice shelf was still unaffected and that a new zone of weakening was not yet established.

The opening of the fracture pair on the ice bridge in 2007 and the fracture formation perpendicular to the ice bridge in 2008, along which the later ice front formed, was best expressed in the ice velocities, both principal stress components, in the first principal strain-rates as well as in-flow strain-rates (Figs. 3&4). From the ice velocities a clear distinction can be made between a western and eastern flow branch on each side of the new fracture pair (Fig. 3e). Increased velocities at the western side indicate the subsequent failure of this part. For the derived quantities, this clear distinction is less evident as they are calculated from spatial derivatives. These fields rather give indications on the restraint or buttressing state. Magnitudes of stress and strain-rate components can be compared to threshold values for crevasse opening (Hambrey and Müller, 1978; Meier, 1958). Such a comparison is certainly intuitive for in-flow strain-rates. The more informative field, however, is the first principal strain as it holds the maximum extension. On the ice bridge in 2007 (Fig. 3g) and just upstream of its remainder in 2008 (Fig. 3k) first principal strain-rates reach values between $\sim 5 \cdot 10^{-5}$ and $1.3 \cdot 10^{-4}$ day$^{-1}$. The narrowest part of the ice bridge holds maximum first principal strain-rates of $\sim 1.2 \cdot 10^{-4}$ day$^{-1}$ in 2008. These values compare to or even exceed threshold strain-rates inferred in areas where crevasses actually opened (Hambrey and Müller, 1978; Meier, 1958).

The first principal stress component also highlights regions, where fractures are likely to open or propagate (Krug et al., 2014; Vaughan, 1993). Threshold values for crevasse opening on various ice shelves fall into a range of 130 and 300 kPa as inferred by Vaughan (1993). For the break-up events in 2007 and 2008 (Figs. 4d, g), inferred stress values fall into this range. Values readily exceed $\sim 200$ kPa on the ice bridge in 2007. In 2008, similar values are found upstream of the ice bridge along the fractures that defined the successive ice-front position (Figs. 4d, g). Values were slightly lower at the narrowest part of the ice bridge, where the ice bridge collapse initiated (Fig. 4g). The interpretation of stress magnitudes is, however, limited as values directly scale with the rate factor choice. The second principal stress component shows some advantage in this respect (Fig. 4b, e, h, k). For all retreat

stages, we confirm that this stress field shows a sign change to positive values in critical regions. Sign changes are scale-free and thus, unaffected by the rate factor choice.

For the interpretation of the second principal strain-rate component (Fig. 3, lowest panels), we follow Doake et al. (1998). They suggested that when the ice front retreats beyond a certain 'compressive arch', the ice-shelf integrity is no longer granted and destabilization is expected. This strain-rate arch separates a seaward area of extension from an inland area of compression. The stress value associated to the 'compressive arch' isoline is not well quantified but "is probably close to the transition from extension to compression" (Doake et al., 1998) and thus, close to the zero isoline of the second principal strain-rate. On WIS, values are generally negative indicating unstable conditions, which is confirmed by general retreat. Yet, the threshold isoline is not clearly specified and might change in time and differ between ice shelves. Moreover, the second principal strain-rate field is found to be not very sensitive to ice-shelf fracturing and gradual ice-front recession, which appears unfavourable when assessing the dynamic state of an ice shelf.

Given the fact that the principal stresses and strain-rates are sufficient quantities for fracture hypothesis, a measure as the stress-flow angle does not lead to additional information for a fracture hypothesis. The distribution in Figure 4i also contradicts the observed consecutive collapse of the ice bridge. In addition, a more extensive retreat upstream of Vere Ice Rise is suggested by the stress-flow angle criterion in the state of 2008. To date, however, the north-eastern ice-front position as shown in 2009 remained unchanged. Further, no threshold value is given, which could serve to clearly delineate critical areas. Hence, the interpretation of stress-flow angles seems delicate und we suggest to use classical measures for fracture hypothesis like principal stress or strain criteria.

**5 Conclusions**

Our study revealed the potential of modern satellite missions to derive a comprehensive overview of dynamic changes on WIS before and during ice-shelf break-up. For this, time-series of ice velocity maps were derived for 1994/96 and 2006-2010. The most significant speed-up occurred in 2008 prior to the final collapse of an ice bridge between Latady and Charcot Island. From velocities alone, we can only speculate that some flow restraint was removed during this event. Derived quantities such as in-flow and first principal strain-rates as well as principal stress components substantially help to identify areas of buttressing or areas prone to fracturing. First principal stress and strain-rates can be compared to previously forwarded threshold estimates for fracture opening on ice-shelves. The inferred stress field scales, however, with the poorly known rate factor, which complicates an interpretation. Independent of this scaling issue, positive values in the second principal surface stress also highlight the regions susceptible to fracturing. Other measures proofed less appropriate to assess the retreat history on WIS. For the stress-flow angle, on the one hand, interpretation is less evident as no threshold is defined below which crevasse opening is notably facilitated. The second principal strain-rate, on the other hand, is mostly negative and thereby explains general retreat. Values are, however, rather insensitive for the detection of ice-shelf fracturing, even for the strong 2008 acceleration. Hence, this measure seems unfavourable when assessing the retreat stages of WIS.

This first assessment of several fields inferred from SAR satellite imagery highlights the great potential of modern satellite missions to better understand the dynamic response of glacial systems, certainly in light of the anticipated climatic changes over Antarctica. However, a comparison of the significance of the various inferred quantities remains limited by the inherent measurement uncertainties. Our assessment could therefore be substantiated by assimilating such time-series with an ice-flow model to simulate equivalent stress and strain fields that are consistent with flow physics, but free of noise from the measurement method.

**Author contributions**

M.R. drafted the design of the study, collected the SAR data and calculated the velocity maps. J.F. calculated strain-rate and stress components as well as stress-flow angles. M.R. and J.F. analysed the results and wrote the manuscript jointly. The figures have been designed by M.R. The manuscript has been revised by M.B. and A.H. This work is embedded in a DFG project initiated by
M.B. and A.H.

**Acknowledgements**

Satellite data was kindly provided by DLR AO mabra_XTI_GLAC0264, ESA AO 4032 and AO 28292. The authors thank the Deutsche Forschungsgemeinschaft (DFG) for support in the framework of the priority program "Antarctic Research with comparative investigations in Arctic ice areas" under grant BR2105/8-1. M.B. received funding by the European Commission
under the 7th Framework Program through the action – IMCONet (FP7 IRSES, action No.319718), J.F. was supported by DFG grant FU1032/1-1. This work was embedded as co-funding activity within the HGF Alliance "Remote Sensing & Earth System Dynamics".

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

**Table 1: Parameter settings used for SAR intensity-offset tracking.**

| Sensor | Sensor wavelength | Tracking window size (range * azimuth) | Step (range/azimuth) |
|---|---|---|---|
| ALOS PALSAR | 23.5 cm L-band | 128*384 | 12/36 |
| ERS-1/2 SAR | 5.6 cm C-band | 256*1280 | 15/75 |

**Table 2: Error estimation of derived velocity fields.**

| Date | Sensor | $\sigma_v^C$ | $\sigma_v^T$ | $\sigma_v$ |
|---|---|---|---|---|
| yyyy-mm-dd--yyyy-mm-dd | | [m/d] | [m/d] | [m/d] |
| 1994-03-02--1994-03-17 | ERS SAR | 0,0645311 | 0,2 | 0,2645311 |
| 2006-05-18--2006-07-03 | ALOS PALSAR | 0,0707297 | 0,03 | 0,1007297 |
| 2006-06-09--2006-07-25 | ALOS PALSAR | 0,249185 | 0,03 | 0,279185 |
| 2006-06-14--2006-07-30 | ALOS PALSAR | 0,267962 | 0,03 | 0,297962 |
| 2007-09-26--2007-11-11 | ALOS PALSAR | 0,25396 | 0,03 | 0,28396 |
| 2007-09-28--2007-11-13 | ALOS PALSAR | 0,1755955 | 0,03 | 0,2055955 |
| 2007-10-20--2007-12-05 | ALOS PALSAR | 0,0428592 | 0,03 | 0,0728592 |
| 2008-09-18--2008-11-03 | ALOS PALSAR | 0,186815 | 0,03 | 0,216815 |
| 2008-09-28--2008-11-13 | ALOS PALSAR | 0,113365 | 0,03 | 0,143365 |
| 2008-10-22--2008-12-07 | ALOS PALSAR | 0,037087 | 0,03 | 0,067087 |
| 2009-09-09--2009-10-25 | ALOS PALSAR | 0,06461115 | 0,03 | 0,09461115 |
| 2009-09-21--2009-11-06 | ALOS PALSAR | 0,0428839 | 0,03 | 0,0728839 |
| 2009-10-01--2009-11-16 | ALOS PALSAR | 0,1476365 | 0,03 | 0,1776365 |
| 2010-08-09--2010-09-24 | ALOS PALSAR | 0,04809755 | 0,03 | 0,07809755 |
| 2010-09-29--2010-11-14 | ALOS PALSAR | 0,0657041 | 0,03 | 0,0957041 |
| 2010-10-09--2010-11-24 | ALOS PALSAR | 0,08 | 0,03 | 0,11 |

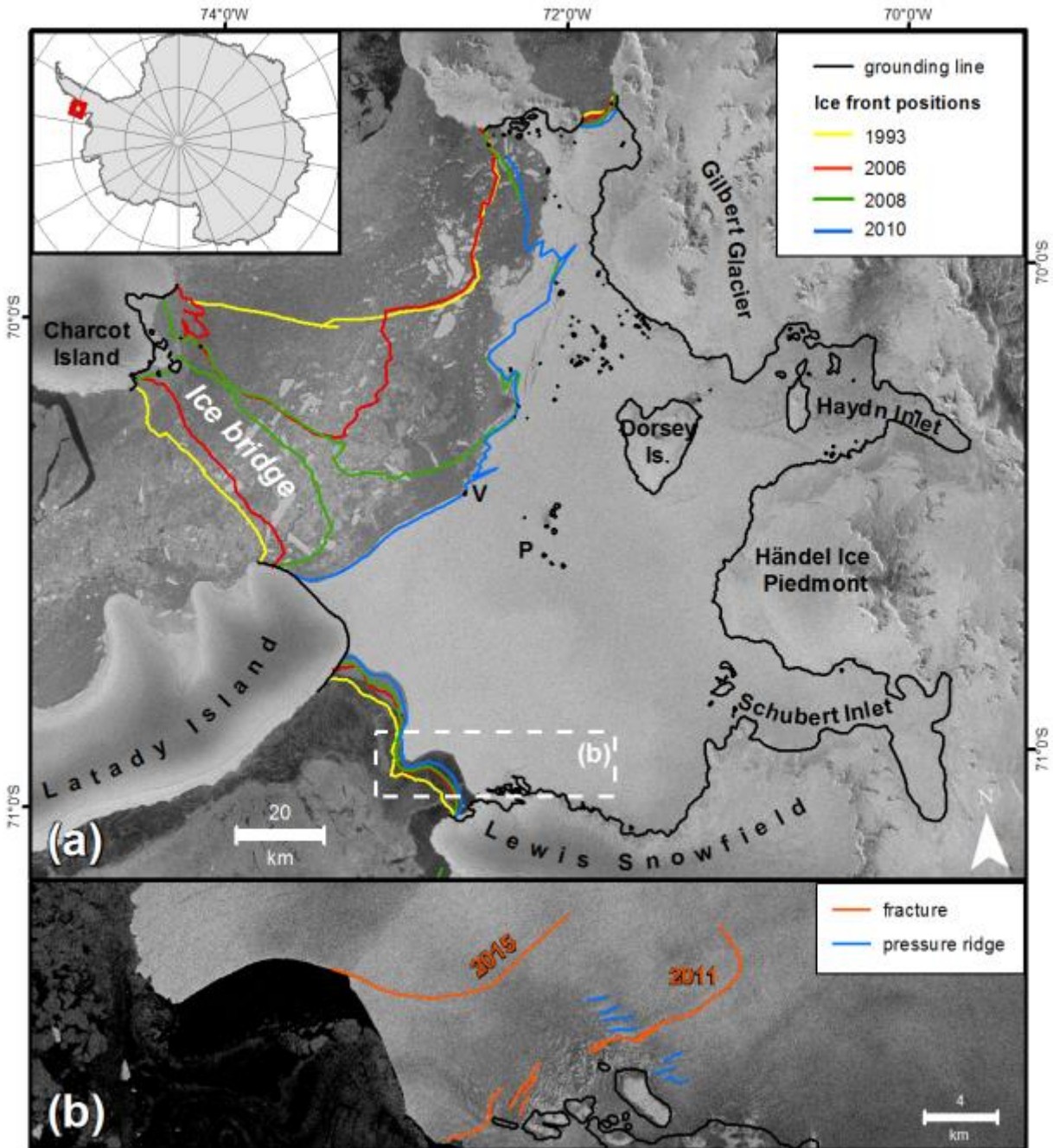

**Figure *1*: (a) Overview map of Wilkins Ice Shelf. The ice-front positions of the years 1993, 2006, 2008 and 2010 as well as the situation in 2015 (background imagery) are shown. The remnant ice connection (ice bridge) between Charcot and Latady Island in its shape from April 2008 is indicated. The location of the grounding line was derived from ERS-1/-2 differential interferometry (supplement material S3). (b) Fracture formation at the south-western ice front as detected in 2011 and 2015 from Envisat ASAR, Sentinel 1a and TanDEM-X imagery. P = Petrie Ice Rises, V = Vere Ice Rise. Background: (a) Sentinel-1a 21-05-2015, (b) Sentinel-1a 06-04-2015 © ESA.**

**Figure** *2*: **Surface velocities of WIS in 1994/96 and between 2006 and 2010 derived from ERS-1/2 (Braun et al., 2009) and ALOS PALSAR intensity-offset tracking. The position of the grounding line in 1995/96 is marked as black line. Panels g) and h) show differences in surface flow between 2008 and 2007, and 2009 and 2008, respectively, for the western part of the ice shelf. P = Petrie Ice Rises, V = Vere Ice Rise. Background imagery: (a) RAMP Mosaic Sep./Oct. 1997 (Jezek, 2013), ERS-1/2 SAR 19-08-1993; (b) Envisat ASAR 25-03-2006; (c) Envisat ASAR 05-08-2007 and 09-09-2007; (d) Envisat ASAR 25-08-2008; (e) Envisat ASAR 22-07-2009; (f) Envisat ASAR 09-12-2010; (g) Envisat ASAR 25-08-2008; (h) Envisat ASAR 22-07-2009, © ESA.**

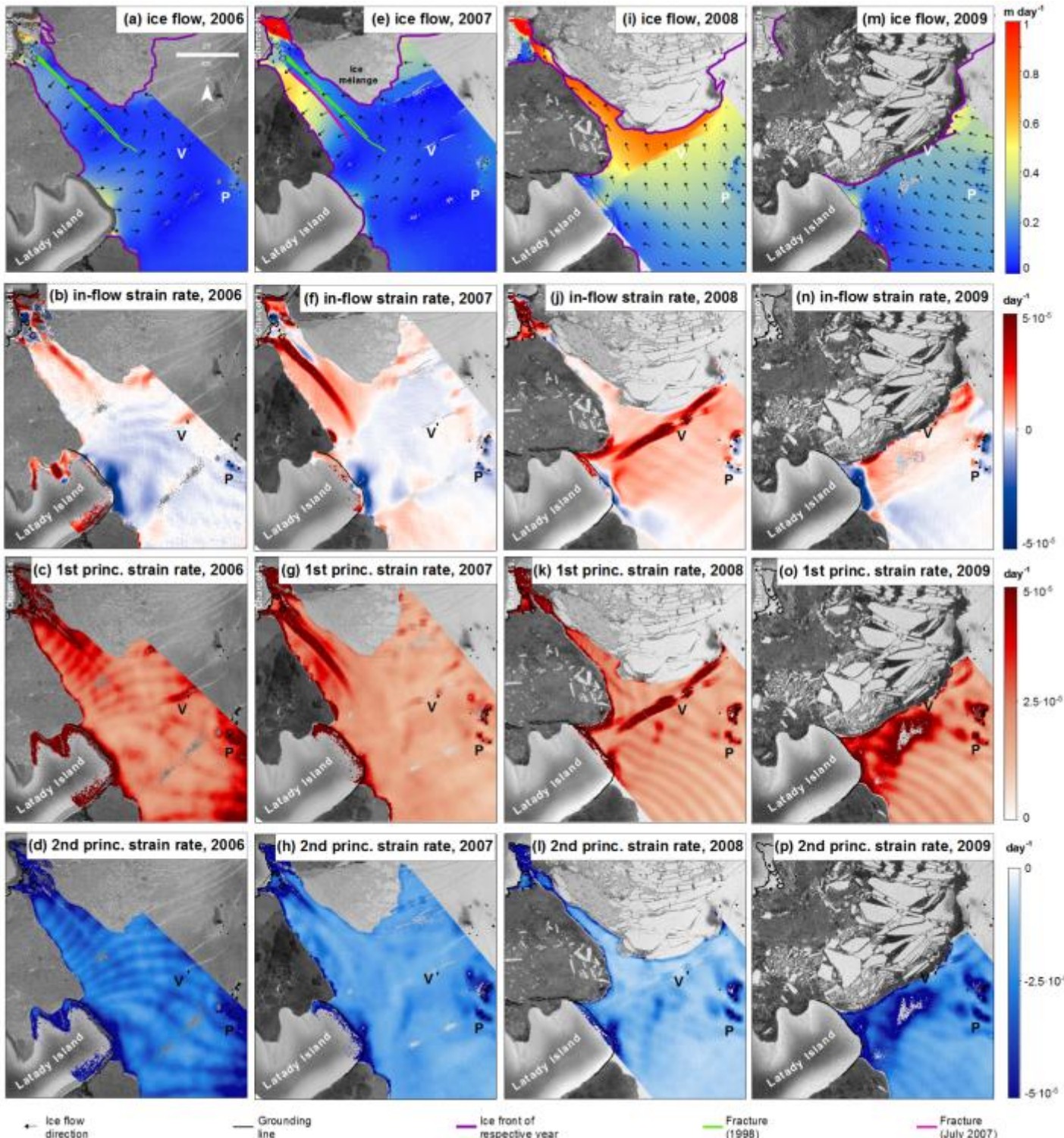

**Figure 3: Surface velocities (first row), strain-rates in flow direction (second row), first and second principal strain-rates for the years 2006-2009 (bottom rows). For depiction, the strain-rate components were filtered using a moving average with a rectangular kernel of 1000m x 1000m. Arrows show the ice-flow direction above a threshold of 0.05 m day⁻¹. For each date the respective ice-front positions are shown. A double fracture system formed in 1998 is shown as green lines in (a) and (e). Another fracture pair developed in July 2007 is marked as purple line in (e). P = Petrie Ice Rises, V = Vere Ice Rise. Background imagery: (a)-(d) Envisat ASAR 25-03-2006; (e)-(h) Envisat ASAR 09-09-2007; (i)-(l) Envisat ASAR 02-08-2008/06-08-2008; (m)-(p) Envisat ASAR 22-07-2009; © ESA.**

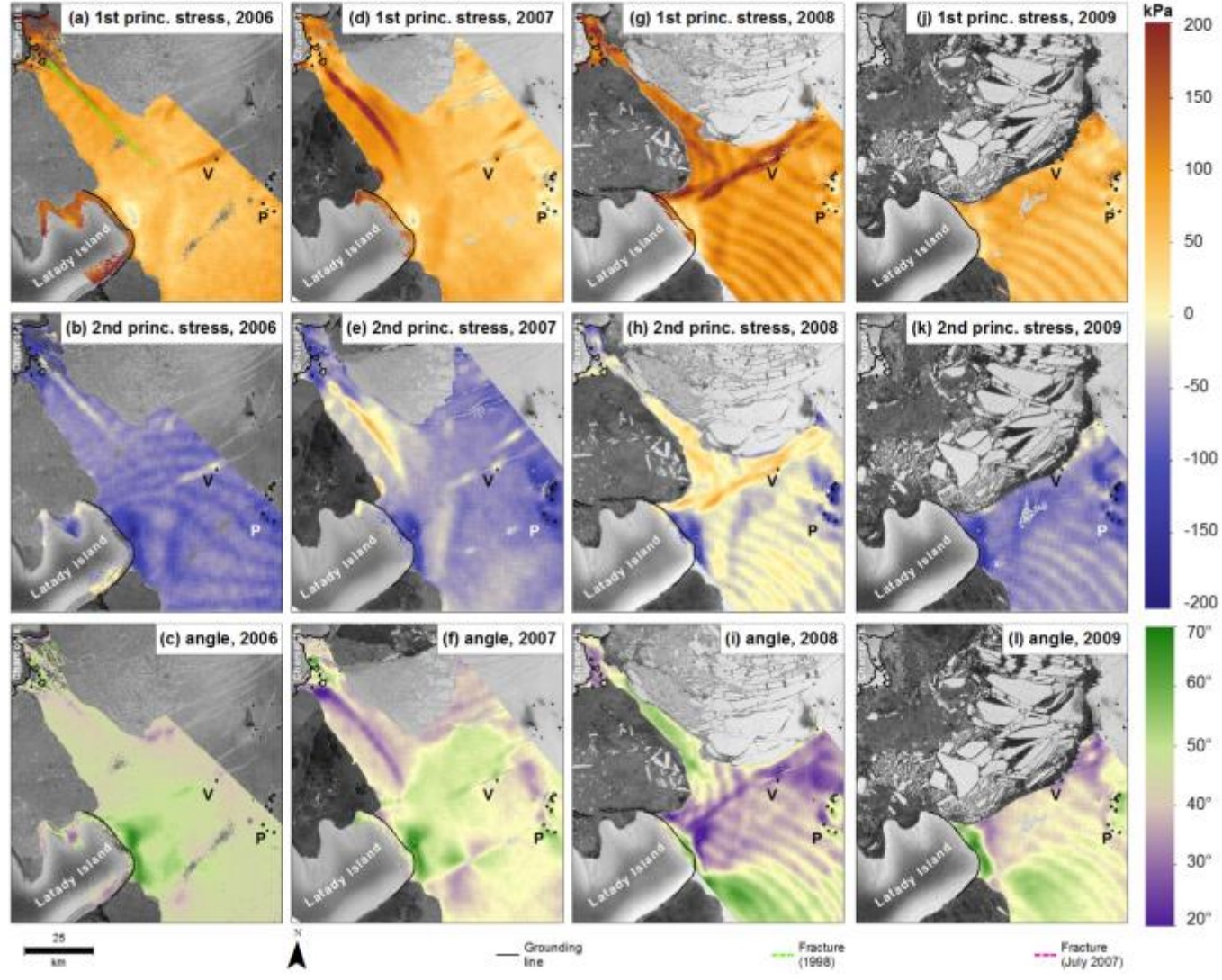

**Figure *4*: First and second principal stress fields as well as stress-flow angles on WIS for the years 2006-2009. P = Petrie Ice Rises, V = Vere Ice Rise. Background imagery: (a)-(c) Envisat ASAR 25-03-2006; (d)-(f) Envisat ASAR 09-09-2007; (g)-(i) Envisat ASAR 02-08-2008/06-08-2008; (j)-(l) Envisat ASAR 22-07-2009; © ESA.**