# Peer review of "Dynamic changes on Wilkins Ice Shelf during the 2006-2009 retreat derived from satellite observations"

_The Cryosphere, 2016_

## Referee Comment (RC1) · RD Drews (Referee) · 15 Nov 2016

**Summary**

Ice shelves buttress ice flowing off the Antarctic continent, and ice-shelf disintegration causes a rapid increase in ice discharge. To predict the future of ice shelves, it is important to derive metrics assessing ice-shelf stability from observations. The Wilkins Ice Shelf (Antarctic Peninsula) has shown significant dynamic changes over the last decades, including thinning and frontal retreat,making it a suitable test case for previously published metrics such as the 'compressive arch' (Doake et al. 1998) or the 'stress-flow angles' (Kulessa et al., 2014).

[Figure]

With this motivation in mind, Rankl et al. present 8 time-slices of surface velocities for the Wilkins Ice Shelf starting in 1994 and ending in 2010. The velocities were derived using intensity/speckle tracking on scenes from various radar satellite sensors. Based on the surface velocities, they derive strain rate fields and the corresponding magnitude/direction of the principal strain rates. Using a simplified rheology (which assumes a spatially/temporally constant rate factor), they also derive the corresponding stress fields with the direction/magnitude of the principal stresses.

The surface velocities quantify temporal changes, in particular an acceleration in 2008 as response to a partial disintegration of an ice bridge between Latady and Charcot Island. Crevasse formation and frontal retreat are tracked in the underlying backscatter images. The derived strain rate and stress fields provide temporally resolved data for testing ice-shelf stability criteria. The authors find that the first principle strain rate (direction of maximum extension) describes observed crevasse opening and the corresponding principal stresses give some indication at which threshold this may occur (subject to the simplified rheology). Change of sign in the second principal stress are independent of the rate factor and help during the interpretation. In particular, positive second principal stresses (marking a purely extensive flow regime) seem to mark areas which are prone to disintegrate. Other measures such as the second strain-rate invariant of the stress-flow angles remain ambiguous.

**General Impression**

The time series of surface velocities provide a strong observational dataset quantifying the dynamic changes of the Wilkins Ice Shelf in the past. I appreciate the derivation of strain rates and stresses (together with their principal components) to understand the underlying mechanisms of ice-shelf stability which is a timely and important topic. Overall I enjoyed reading this manuscript and from my point of view, this paper will fulfill the standards of The Cryosphere. Below I do suggest some revisions which should be addressed and which hopefully will help to improve the final version of the paper. I hope the authors persever to go through my comments, and to turn this already well-

developed TCD paper into a nice TC publication.

Reinhard Drews

**General Comments**

1. The paper is in many places unnecessarily descriptive where it could be rigorous. This is manifested in expressions such as "some of the measures [..] emerge to be more or less applicable" p.7/l.29 (is it applicable or not, and why?), "..acceleration is somewhat lower" (how much?), "..showed very small velocities in March.." (how slow and slow compared to what?). Although the individual examples are all minor, the repeated occurrence of these type of descriptions makes the paper imprecise and speculative in places. Below I mention a detailed list and suggest improvements.

2. Error estimates of the velocities are derived in the supplements, but I wonder if these errors are complete. From my experience with intensity tracking, it is (at least sometimes) required to calibrate the data (e.g. offset it at rock outcrops) and even the calibrated fields may show non-zero, spatially varying values in the difference fields of mosaics (due to errors in coregistration, orbital information, atmospheric contribution,...). In Fig. 2b,c,d,e cutting edges are visible, but the authors do not report their magnitude and attribute them to monthly flow variations. With the information available in the paper, this appears overinterpreted and should be better justified (how were the data calibrated, what is the magnitude of the cutting edges and what other evidence exists for monthly flow variations?). Along those lines, I suggest to show the full difference field and not restrict it to the western part only.

3. The wave-like pattern which appears in the strain rates (e.g. Fig. 2 c,k,o) and elsewhere is quite prominent but is only briefly mentioned in an isolated sentence (p. 6/l. 12) when describing the stresses. This must be mentioned earlier (e.g. section 3.1) and the explanation should be expanded. How is it possible that the tracking algorithm creates such a pattern, and if so, why only in the scenes from 2008 and 2009? Ionospheric path delays are possible but may not be the only option. Somewhat

immodestly I refer to Drews et al. 2009 (IEEE Geosc. Rem. Sens. Fig. 4) where a similar wave-like pattern has been linked to variations in troposhperic water vapor content. However, this was using differential SAR interferometry and I don't know how this would appear in flow fields based on intensity/speckle tracking.

4. I have no prior experience in applying the ice-shelf stability criteria mentioned in the manuscript and I enjoyed learning more about them. However, in the end I was left with a foggy impression which criteria are applicable where and under which conditions. A good example for this confusion is the paragraph p.7/l. 38f stating that the authors "want to follow Doak et al. (1998)" in saying that "when the ice front retreats beyond a certain isoline [in the second principal strain rates] destabilization is expected". However, two sentences later is is stated that the observed "second principal strain rates are [..] insensitive during the retreat of WIS" which makes me think that the Doak-criteria does not fit because WIS has retreated. However, the authors go on stating that the "negative values suggest general recession" which again is what has been observed. I am sure that I misunderstand something here, but even after reading this paragraph multiple times I am still left in the dark whether or not the second principal strain rates are a good or a bad metric for assessing the stability in this case. Similar ambiguities also occur at other places mentioned below. Summarizing, I am not questioning the analysis, but I suggest that the authors put more effort into clearly stating their findings which are currently partially hidden. This also includes sharpening the conclusions. Detailed comments are mentioned below.

**Specific Comments**

p1 l 12 how about "A total area of $2135 \pm 75 km^2 (XX percent of the total ice shelf area)$

p1 l 12 "multi-temporal": how about "in order to derive temporal variability of surface flow

p1 l 14 "which were forwarded" – > "which were forwarded previously"

p1 l 21 "pins down" –> "defines"

p1 l 22 remove "general"

p1 l 23 ice-shelf "disintegration" "collapse" "break-up" are used interchangeably. Maybe better stick to one term if you mean the same thing to avoid confusion for the reader. Same for ice-front "retreat" "recession"

p1 l 26 I have difficulties imagining a tongues of a tributary glacier feeding into an ice shelf. Can you clarify?

p1 l 33 I supposed "within only one week" refers to the time period of ice-shelf disintegration. It could also mean that the criterion was published on week after the breakup. Rephrase.

p2 l 9: "under" –> "during"

p2 l 26: "multi-temporal" –> "time series"

p2 l 35: I suggest to mention also normalized values of the area (e.g. relative to the entire ice-shelf extent in year XX), because it is hard to get a feel for these numbers otherwise.

p2 l 38: Surface melt ponding may still play a role in ice-shelf disintegration. Stick a "exclusively" in there or restrict this sentence to the WIS. Otherwise you neglect the role of surface-melt ponding in general, which I think is not what you intend.

p3 l 19: Compared to which reference direction are flow vectors filtered? Where does this direction come from?

p3 l 16: specify what you mean with "relative orbits" as opposed to just "orbits".

p3 l 21f: I get the feeling that "relative" refers to uncalibrated data where as "absolute" refers to calibrated data. Please specify how the calibration was done (on rock outcrops?) and be more clear about using the words relative/absolute

p3 l 5 : What is the final gridding of the velocities and did you apply any calibration?

p3 l 25: I am not yet convinced that the deviations in the overlapping parts of the mosaicked surface flow are solely due to monthly ice-flow variations. Are there any independent data showing that ice-flow varies on monthly time scales and by how much? How can you be sure that the observed offsets are not due to the processing artefacts (i.e. unresolved coregistration offsets, imprecise orbital information or maybe even atmospheric contributions)? I think this point should be discussed in more detail, because the difference field is a major result of this paper. Currently I have the feeling that the overall data accuracy of the velocities is overestimated (although Fig. 2g is and will remain significant). I am happy to be shown otherwise.

p3 l 25: Given that TC has no rigid page limit, I don't see the advantage of having the error analysis (which I find important) in the supplements.

p4 l 11: Pattyn 2003 is a paper about including higher-order mechanics in ice-flow models. However, the use of deviatoric stresses is more general and already occurs in the shallow ice approximation. Therefore, I find this reference a little misleading and suggest a textbook reference (Greve Blatter, Cuffey Patterson,...)

p4 l 12 "data are acquired" (data are plural)

p4 l 20 remove "In addition,"

p4 l 20 "necessarily linear" –> "is non-linear" (I consider this as the default in glaciology. The "not necessarily linear" somehow implies otherwise.)

p4 l 25 Remove "generally". The rate factor always depends on temperature.

p4 l 30 Not sure if it is fully correct to state the "stress tensor" is symmetric by construction. The symmetry reflects conservation of angular momentum. I don't want to be witty here, just point out small imprecise language. Maybe rephrase.

p4 l 34 E-5 (also at other places) is not TC style

p5 l 5f: I am more used to have velocities in meters per year but certainly don't insist on it.

p5 l 11: Quantify "very small" because velocities are a main result of this paper there is no need to be descriptive here. Same holds for terms like "slightly higher" etc.

p5 l 13 "large double fracture" how large?

p5 l 17 "the consecutive acceleration was most expressed" how about "the consecutive acceleration was largest at.."

p5 l 18 "0.8 meters per day" is a velocity, not an acceleration. Do you mean velocity increase? Again, mentioning relative values (e.g. and XX percent increase) would help to better grasp the meaning of these numbers.

p5 l 19 terms like "somewhat lower" are unnecessarily descriptive for primary results. Same for "rather stagnant".

p5 l 26: How about: "We calculate strain rates at each point in a local coordinate system with the two axis aligned parallel and perpendicular to the local flow direction."

p5 l 26: Mention and check that the invariants (principal strain rates etc.) are indeed invariant compared to strain rates which are calculated in a global coordinate system.

p5 l 30: Which threshold was chosen and why? Fig. 3 mentions 0.05 m/d.

p5 l 33: "slight inflow"?

p6 l 2: a "certain direction" is imprecise. It is along the eigenvector corresponding to the eigenvalue.

p6 l 2: As a non-sailor I constantly mix up "lee" and "luv". Also people may understand this term from a perspective of atmospheric circulation. I think it is better to talk about "upstream" and "downstream" in these cases (and elsewhere).

p6 l 6: quantify "less negative"

p6 l 14: This must be mentioned earlier and discussed in more detail (see general comment above).

p6 l 33 remove "more or less"

p6 l 35 when interpreting strain fields for fracture formation it is important to explicitly state the resolution of velocities and strain rates.

p7 l 16 "Even if zero is not passed, "–> "Even if values remain negative, ..."

p7 l 29 "more or less" is not helpful here. Better write a clear sentence stating that no single stress/strain field can fully explain the observations (or something like that).

p7 l 30 I suggest avoiding "might be" (and "could be" later on) and instead clearly stating the uncertainties that are indicated with these formulations.

p7 l 36 I do not grasp the meaning of this paragraph and suggest rephrasing (see general comment above).

p7 l 38 "certain" isoline? Is it not the 0 isoline specifying pure extension in all directions (as stated in intro)?

p7 l 38 "retreats/recedes" what is the difference between retreating and receding and why are both verbs mentioned here?

p8 l 9 This paragraph of fractures fits to the one started in p7 l29. I also suggest to move the last paragraph of the conclusions to this section to have all fracture-related things together.

p 8 l 29: This paragraph brings up a observations which have not been discussed previously in the paper. I suggest not to introduce novel things in the conclusions (see previous comment).

p8 l 16 High variability or high uncertain (because flow direction is ill-constrainedy?

Figure 1

I don't see the red nor the cyan line. On the other hand I see an orange ice-front position not mentioned in legend. Something went wrong here.
* * *

---

## Referee Comment (RC2) · T.ÂăA. Scambos (Referee) · 22 Nov 2016

T.ÂăA. Scambos (Referee)

teds@nsidc.org

Using a time-series of InSAR and speckle-tracked radar images of velocity, Rankl et al. present very good detailed study of the events occurring on the Wilkins Ice Shelf spanning the 1990s and through the series of major calvings and disintegrations occurring in 2008 and 2009. The focus onf the paper is on ice flow and strain rates as the ice shelf evolves. This analysis of events provide insight into how stresses are transferred within ice shelves, and illustrate the power of good sequential ice velocity data in diagnosing causality for riftings and calvings.

In general, the writing could be tighter. There are some odd constructions for an English first-language reader, although the meaning is clear enough. But some work on the text

could probably reduce the length by 10% and make it an easier, more efficient read. In reading it, I was interested by the main data figures (Fig2, 3, and 4) but found the discussion hard to follow because of the detail – all quite accurate, but it seemed to go slowly through this part when it could have been more interesting.

My main comment concerns the interpretations of Figure 2 and 3 and 4 – the data look very good, and they provide a clear story – although I see you are cautioning about data quality in slow-moving areas. What is striking is the abrupt shift in the pattern after the calving and disintegrations of Feb-March, and especially after July, 2008 – this shows that the middle section of the ice bridge was an important buttress, and that the last ice bridge whisker was already nearly detached at the northern end (see Fig 5 in Braun et al. 2009 and Fig2 in Scambos et al. 2009 – the connection to Charcot Is. is rifted and sheared prior to the removal of the central ice bridge piece). With the loss of the middle ice bridge section, strong extensional stress is present just north of the Vere ice rise, and the ice soon rifts away .. Sections 4.1 and 4.2 work through the data in the figures slowly... it would be more emphatice and clear to introduce the three figures briefly (just say what they are) and then discuss the evolution of the fractures and strain rates in a more story-like fashion. Readers would retain the events and significance better.

Or perhaps open with a) brief description of the data shown in the figures, and caveats, then, b) an overview 'story' of how events proceeded and the major fractures and shifts in strain patterns, and then c) perhaps some kind of review of the details captured by the Fig3 and 4 data that you are discussing on Page 6 and 7.

I think the data make the sudden rearrangement the highlight of the paper; they underscore the relative lack of importance of the easternmost section of the ice bridge (or, if you like, the importance of the middle section.)

Overall, I think the manuscript is nearly publishable as it is, but would benefit and be more likely to be remembered and cited with another round of editing with respect to telling the story more clearly and succinctly. The conclusions have this kind of 'voice'.

P2L1 – in several papers, I've been trying to reserve this word for the kind of fine-scale rapid calving that was observed on Larsen B in January - March 2002 and Wilkins in Feb29-March8 2008. Please use the word 'collapse' here, since the ice shelf instability caused by the loss of the compressive arch might simply result in a series of large-scale calvings spanning months or even years, and not a true 'disintegration'.

P3L13 SNR: This needs a bit of unpacking – what you mean is a correlation peak height that is less than 4 times the mean correlation away from the peak. 'Signal' to 'noise' is a bit obscure here.

P5L33-34 Yes, extensional strain, but once the rift had formed in July 2007, almost all of the 'strain' would be taken up by the rift widening. It's what the strain rate was prior to the new rift (which, agreed, would have been formed by the stress build-up).

P7L3 – '…it is obvious ….. ' this phrase is odd, the block might have just calved away intact?

P7L8-25 I find this section somewhat of a difficult read – too tentative, to qualified; there's a basic story from the data, but it's obscured by nuance here. For example, L19-21, the ice bridge has a stabilizing effect, yes, and that places it under compressive strain along its axis, thus leading to, not failing to prevent, its eventual collapse - ?

Fig3 and Fig4 – there is a white (fig3) and red (fig4) dot near the northeastern corner of the ice bridge – what does that signifiy? Please describe it in the captions.

Please also note the supplement to this comment:
http://www.the-cryosphere-discuss.net/tc-2016-218/tc-2016-218-RC2-supplement.pdf

[Figure]

**Supplement:**

[revised manuscript text omitted]

---

## Author Comment (AC1) · 20 Jan 2017

**Point-by-point reply to review comments on 'Dynamic changes on Wilkins Ice Shelf during the 2006-2009 retreat derived from satellite observations' by Rankl et al.**

The authors want to thank both reviewers for their careful reading of the manuscript. The comments were very useful and helped to improve the manuscript substantially. Large parts of the manuscript have been re-structured following the suggestions given by Ted Scambos. More methodological aspects brought up by Reinhard Drews have been addressed extensively in the revised version of the manuscript. In the following, all comments are addressed specifically and changes in the manuscript indicated. We hope, both reviewers support the changes made in the revised manuscript.

**Comments by Reinhard Drews**

Summary
Ice shelves buttress ice flowing off the Antarctic continent, and ice-shelf disintegration causes a rapid increase in ice discharge. To predict the future of ice shelves, it is important to derive metrics assessing ice-shelf stability from observations. The Wilkins Ice Shelf (Antarctic Peninsula) has shown significant dynamic changes over the last decades, including thinning and frontal retreat, making it a suitable test case for previously published metrics such as the 'compressive arch' (Doake et al. 1998) or the 'stress-flow angles' (Kulessa et al., 2014).

With this motivation in mind, Rankl et al. present 8 time-slices of surface velocities for the Wilkins Ice Shelf starting in 1994 and ending in 2010. The velocities were derived using intensity/speckle tracking on scenes from various radar satellite sensors. Based on the surface velocities, they derive strain rate fields and the corresponding magnitude/direction of the principal strain rates. Using a simplified rheology (which assumes a spatially/temporally constant rate factor), they also derive the corresponding stress fields with the direction/magnitude of the principal stresses. The surface velocities quantify temporal changes, in particular an acceleration in 2008 as response to a partial disintegration of an ice bridge between Latady and Charcot Island. Crevasse formation and frontal retreat are tracked in the underlying backscatter images. The derived strain rate and stress fields provide temporally resolved data for testing ice-shelf stability criteria. The authors find that the first principle strain rate (direction of maximum extension) describes observed crevasse opening and the corresponding principal stresses give some indication at which threshold this may occur (subject to the simplified rheology). Change of sign in the second principal stress are independent of the rate factor and help during the interpretation. In particular, positive second principal stresses (marking a purely extensive flow regime) seem to mark areas which are prone to disintegrate. Other measures such as the second strain-rate invariant of the stress-flow angles remain ambiguous.

General Impression
The time series of surface velocities provide a strong observational dataset quantifying the dynamic changes of the Wilkins Ice Shelf in the past. I appreciate the derivation of strain rates and stresses (together with their principal components) to understand the underlying mechanisms of ice-shelf stability which is a timely and important topic. Overall I enjoyed reading this manuscript and from my point of view, this paper will fulfill the standards of The Cryosphere. Below I do suggest some revisions which should be addressed and which hopefully will help to improve the final version of the paper. I hope the authors persever to go through my comments, and to turn this already well-developed TCD paper into a nice TC publication.
Reinhard Drews

**General Comments**

1. The paper is in many places unnecessarily descriptive where it could be rigorous. This is manifested in expressions such as "some of the measures [..] emerge to be more or less applicable" p.7/l.29 (is it applicable or not, and why?), "..acceleration is somewhat lower" (how much?), "..showed very small velocities in March.." (how slow and slow compared to what?). Although the individual examples are all minor, the repeated

occurrence of these type of descriptions makes the paper imprecise and speculative in places. Below I mention a detailed list and suggest improvements.

Ted Scambos asked for a profound re-organization of the Results section. During the rewriting, we tried to remove many of these imprecise and ambiguous formulations.

2. Error estimates of the velocities are derived in the supplements, but I wonder if these errors are complete. From my experience with intensity tracking, it is (at least sometimes) required to calibrate the data (e.g. offset it at rock outcrops) and even the calibrated fields may show non-zero, spatially varying values in the difference fields of mosaics (due to errors in coregistration, orbital information, atmospheric contribution,...).

We thank the reviewer for this comment. The reviewer is right that additional calibration of the velocity fields might be useful in some cases. We did not do any additional subtraction of measured offsets on stable ground as the reviewer suggests. However, we think that the full possible error contribution is expressed in the term $\sigma_v^C$. This value captures all errors related to the co-registration procedure and is hence, larger than an error that would have been assessed after additional calibration. Further error contribution related to the orbital information or the atmospheric influence is hard to estimate, however, we would welcome any suggestions to do so. We think the error estimation over non-moving areas as a reference is a standard procedure when using intensity-offset tracking methods (Burgess et al., 2012; McNabb et al., 2012; Quincey et al., 2009, 2011; Seehaus et al., 2015). The second value $\sigma_v^T$, which adds to the error calculation considers further error sources related to the tracking algorithm, the spatial resolution and time interval of image acquisitions and, in our opinion, improves the overall error estimation. In a revised manuscript, we will address the error computations in more detail as suggested by the reviewer.

In Fig. 2b,c,d,e cutting edges are visible, but the authors do not report their magnitude and attribute them to monthly flow variations. With the information available in the paper, this appears overinterpreted and should be better justified (how were the data calibrated, what is the magnitude of the cutting edges and what other evidence exists for monthly flow variations?). Along those lines, I suggest to show the full difference field and not restrict it to the western part only.

During the preparation of the velocity results, we first tried to calibrate the single velocity fields to each other. However, due to the mostly structureless ice shelf with in some parts very few non-moving areas, only limited overlapping areas could be identified to be used for calibration. The offsets between tracks are also not linear (range between 3 and 18 m/yr) and hence, a subtraction is not a straight forward approach without sufficient reference on stable ground. Our efforts did not lead to substantially more consistent velocity fields when also considering the offsets on stable ground. Since there are no in-situ measurements on WIS and only limited satellite data coverage suitable for intensity-offset tracking or interferometry, we cannot give more details on short-term velocity variations. In addition, this is also the reason for varying co-registration accuracies during offset racking. Hence, we decided against further calibration and to show the 'as is' flow fields, which might show short-term flow variations or, as the reviewer correctly pointed out, processing artefacts. In a revised version of the paper, we will address this point more clearly and explain the reasons for our approach and give values of the offsets. We also decided against any coarse smoothing techniques as often applied to large scale ice sheet velocities, in order to keep as much detail as possible. The cutting edges are also the reason, why we have decided to show flow differences for the western part of the ice shelf only. This part of the ice shelf is most important for the aim of our paper, since dynamic changes due to frontal break-up are obvious in this part, whereas the eastern part of the ice shelf is not influenced mainly due to the blocking effect of numerous ice rises.

3. The wave-like pattern which appears in the strain rates (e.g. Fig. 2 c,k,o) and elsewhere is quite prominent but is only briefly mentioned in an isolated sentence (p. 6/l. 12) when describing the stresses. This must be mentioned earlier (e.g. section 3.1) and the explanation should be expanded. How is it possible that the tracking algorithm creates such a pattern, and if so, why only in the scenes from 2008 and 2009? Ionospheric path delays are possible but may not be the only option. Somewhat immodestly I refer to Drews et al. 2009 (IEEE Geosc. Rem. Sens. Fig. 4) where a similar wave-like pattern has been linked to variations in troposhperic water vapor content. However, this was using differential SAR interferometry and I don't know how this would appear in flow fields based on intensity/speckle tracking.

We thank the reviewer for this advice and will discuss this in more detail in an updated manuscript. We also agree that the tracking algorithm is the most unlikely source for these patterns due to the reasons mentioned by the reviewer. Other publications have found comparable patterns in ice velocities derived from intensity tracking and related them to fluctuations in the polar ionospheric electron density (e.g., Gray et al., 2000; Joughin, 2002; Nagler et al., 2015). These fluctuations affect not only the interferometric phase, but also a mismapping of the azimuth pixel position (Gray et al., 2000). Especially slow moving areas are affected by these patterns, which is true for the WIS. The proposed wavelength of the pattern of 5-10 km is applicable to WIS (Gray et al., 2000). An improvement by averaging the flow fields over multiple acquisitions as proposed in Nagler et al. (2015) is not possible in our case due to lack of further, suitable image pairs. Solar activity is strongly variable in time and would hence explain why we observe the pattern only on a few image pairs.

We are thankful about the advice mentioning the possible influence of the tropospheric water vapor content. This is certainly another temporarily variable source of error. However, related studies have mentioned an influence on C-Band InSAR only (Williams et al., 1998), which is a more sensitive technique on the path delays due to water vapor. We will include this as another possible reason for the observed pattern, although we did not find further information about any effects on the results derived from intensity tracking. In our view, ionospheric disturbances remain the most likely reason for the observed patterns.

4. I have no prior experience in applying the ice-shelf stability criteria mentioned in the manuscript and I enjoyed learning more about them. However, in the end I was left with a foggy impression which criteria are applicable where and under which conditions. A good example for this confusion is the paragraph p.7/l. 38f stating that the authors "want to follow Doak et al. (1998)" in saying that "when the ice front retreats beyond a certain isoline [in the second principal strain rates] destabilization is expected". However, two sentences later it is stated that the observed "second principal strain rates are [..] insensitive during the retreat of WIS" which makes me think that the Doak-criteria does not fit because WIS has retreated. However, the authors go on stating that the "negative values suggest general recession" which again is what has been observed. I am sure that I misunderstand something here, but even after reading this paragraph multiple times I am still left in the dark whether or not the second principal strain rates are a good or a bad metric for assessing the stability in this case. Similar ambiguities also occur at other places mentioned below. Summarizing, I am not questioning the analysis, but I suggest that the authors put more effort into clearly stating their findings which are currently partially hidden. This also includes sharpening the conclusions. Detailed comments are mentioned below.

The reviewer's comment again points out imprecise formulations, which we hopefully could improve by largely rewriting the Results section. The specific section now reads as follows:

*'For the interpretation of the second principal strain-rate component (Fig. 3, lowest panels), we want to follow Doake et al. (1998). They forwarded that when the ice front retreats beyond a certain 'compressive arch', the ice-shelf integrity is no longer granted and destabilization is expected. This strain-rate arch separates a seaward area of extension from an inland area of compression. The stress value associated to the 'compressive arch' isoline is not well quantified but "is probably close to the transition from extension to compression" (Doake et al., 1998) and thus, close to the zero isoline of the second principal strain-rate. On WIS, values are generally negative indicating unstable conditions, which is confirmed by general retreat. Yet, the threshold isoline is not clearly specified and might change in time and differ between ice shelves. Moreover, the second principal strain-rate field is found to be not very sensitive to ice-shelf fracturing and gradual ice-front recession, which appears unfavourable when assessing the dynamic state of an ice shelf.'*

**Specific Comments**

p1 l 12 how about "A total area of 2135±75km 2 (Xxpercent of the total ice shelf area)

The corresponding value has been added to the text.

p1 l 12 "multi-temporal": how about "in order to derive temporal variability of surface flow

The sentence has been changed into:

*'The present study uses time-series of SAR satellite observations (1994/96, 2006-2010) in oder to derive variations in surface flow speed from intensity offset tracking methods.'*

p1 l 14 "which were forwarded" – > "which were forwarded previously"

This sentence has been removed during rewriting.

p1 l 21 "pins down" –> "defines"

This sentence has been removed.

p1 l 22 remove "general"

This sentence has been removed.

p1 l 23 ice-shelf "disintegration" "collapse" "break-up" are used interchangeably. Maybe better stick to one term if you mean the same thing to avoid confusion for the reader. Same for ice-front "retreat" "recession"

We tried to clear the terminology in the text and hope it is less confusing now.

p1 l 26 I have difficulties imagining a tongues of a tributary glacier feeding into an ice shelf. Can you clarify?

We are not really sure what the reviewer is referring to.

p1 l 33 I supposed "within only one week" refers to the time period of ice-shelf disintegration. It could also mean that the criterion was published on week after the breakup. Rephrase.

This sentence has been changed accordingly.

p2 l 9: "under" –> "during"

This sentence has been changed accordingly.

p2 l 26: "multi-temporal" –> "time series"

This sentence has been changed accordingly.

p2 l 35: I suggest to mention also normalized values of the area (e.g. relative to the entire ice-shelf extent in year XX), because it is hard to get a feel for these numbers otherwise.

The corresponding value has been added to the text.

p2 l 38: Surface melt ponding may still play a role in ice-shelf disintegration. Stick a "exclusively" in there or restrict this sentence to the WIS. Otherwise you neglect the role of surface-melt ponding in general, which I think is not what you intend.

This sentence has been changed accordingly.

p3 l 19: Compared to which reference direction are flow vectors filtered? Where does this direction come from?

As proposed in Burgess et al. (2012) the offset fields were filtered for unreasonable values by using an efficient algorithm that compares the orientation and magnitude of a displacement vector in respect to its surrounding vectors. In terms of flow direction this means, that each flow vector in a predefined moving window (here: 5x5 pixels) is compared to the orientation assigned to the window's center pixel. For this, thresholds of 20, 18 and 12 degrees were chosen (Burgess et al., 2012). In an iterative process, each vector which deviates from the defined threshold was deleted.

p3 l 16: specify what you mean with "relative orbits" as opposed to just "orbits".

A relative orbit can be compared to e.g. the Landsat path number. It describes the repeating coverage of a satellite. For better understanding we changed the manuscript and used another term for this (applies also to next comment).

p3 l 21f: I get the feeling that "relative" refers to uncalibrated data where as "absolute" refers to calibrated data. Please specify how the calibration was done (on rock outcrops?) and be more clear about using the words relative/absolute

The authors apologize for the misleading phrasing. The text referred to the satellite's repeating flight path (relative), and the absolute magnitude of the derived flow fields (compared to relative magnitudes derived from e.g. InSAR). This section was changed for better understanding.

p3 l 5 : What is the final gridding of the velocities and did you apply any calibration?

The final gridding of the velocity fields is 50x50 m. This information has been added to the text. The co-registration procedure relies on the masked intensity images, i.e. non-moving areas such as bedrock and ice rises. Hence, we did not apply any further calibration of the velocity fields, since the co-registration accuracy was at a high level.

p3 l 25: I am not yet convinced that the deviations in the overlapping parts of the mosaicked surface flow are solely due to monthly ice-flow variations. Are there any independent data showing that ice-flow varies on monthly time scales and by how much? How can you be sure that the observed offsets are not due to the processing artefacts (i.e. unresolved coregistration offsets, imprecise orbital information or maybe even atmospheric contributions)? I think this point should be discussed in more detail, because the difference field is a major result of this paper. Currently I have the feeling that the overall data accuracy of the velocities is overestimated (although Fig. 2g is and will remain significant). I am happy to be shown otherwise.

See comment above.

p3 l 25: Given that TC has no rigid page limit, I don't see the advantage of having the error analysis (which I find important) in the supplements.

The description of the error estimation has been added to the manuscript.

p4 l 11: Pattyn 2003 is a paper about including higher-order mechanics in ice-flow models. However, the use of deviatoric stresses is more general and already occurs in the shallow ice approximation. Therefore, I find this reference a little misleading and suggest a textbook reference (Greve Blatter, Cuffey Patterson,…)

We changed the reference accordingly.

p4 l 12 "data are acquired" (data are plural)

The sentence has been changed accordingly.

p4 l 20 remove "In addition,"

The sentence has been changed accordingly.

p4 l 20 "necessarily linear" –> "is non-linear" (I consider this as the default in glaciology. The "not necessarily linear" somehow implies otherwise.)

The sentence has been corrected as suggested.

p4 l 25 Remove "generally". The rate factor always depends on temperature.

The sentence has been changed accordingly.

p4 l 30 Not sure if it is fully correct to state the "stress tensor" is symmetric by construction. The symmetry reflects conservation of angular momentum. I don't want to be witty here, just point out small imprecise language. Maybe rephrase.

The reviewer's comment points at a small detail which we certainly want to accommodate. The passage reads now as follows:

*'Strain and stress components are real and respective tensors are symmetric which ultimately reflects the conservation of angular momentum in continuum mechanics.'*

p4 l 34 E-5 (also at other places) is not TC style

The notation has been changed throughout the manuscript.

p5 l 5f: I am more used to have velocities in meters per year but certainly don't insist on it.

We would like to keep the velocity indications in meter per year and hope the reviewer is ok with this.

p5 l 11: Quantify "very small" because velocities are a main result of this paper there is no need to be descriptive here. Same holds for terms like "slightly higher" etc.

The sentence has been changed accordingly.

p5 l 13 "large double fracture" how large?

The fracture pair was about 45km long. This information was added to the text.

p5 l 17 "the consecutive acceleration was most expressed" how about "the consecutive acceleration was largest at.."

The sentence has been changed accordingly.

p5 l 18 "0.8 meters per day" is a velocity, not an acceleration. Do you mean velocity increase? Again, mentioning relative values (e.g. and XX percent increase) would help to better grasp the meaning of these numbers.

The sentence has been changed accordingly. The velocity increase nearly tripled at the narrowest part of the ice bridge. We hope the sentence is clearer now.

p5 l 19 terms like "somewhat lower" are unnecessarily descriptive for primary results. Same for "rather stagnant".

The first sentence mentioned here has been removed from the manuscript. Also, we deleted 'rather'.

p5 l 26: How about: "We calculate strain rates at each point in a local coordinate system with the two axis aligned parallel and perpendicular to the local flow direction."

Corrected following the reviewer's suggestion. We hope this clarifies how the in-flow strain rates are computed.

p5 l 26: Mention and check that the invariants (principal strain rates etc.) are indeed invariant compared to strain rates which are calculated in a global coordinate system.

The decomposition was already cross-checked by verifying that first principal components are larger than second principal components. For the strain-rates we additionally checked whether the values in flow direction fall between the minimum and maximum defined by the first and principal components.

p5 l 30: Which threshold was chosen and why? Fig. 3 mentions 0.05 m/d.

Yes, a threshold of 0.05 m/day was chosen here. This information has been added to the text here, we thank the reviewer for his careful reading. This threshold has been chosen based on the visual interpretation of the ice flow during image analysis. Flow directions in areas of magnitudes lower than 0.05 m/day weren't thoroughly convincing.

p5 l 33: "slight inflow"?

The corresponding sentence has been changed into:

*'Additional inflow originating from Latady Island leads to compressive strain-rates further south.'*

p6 l 2: a "certain direction" is imprecise. It is along the eigenvector corresponding to the eigenvalue.

This somewhat imprecise formulation was removed during re-structuring of the Results section.

p6 l 2: As a non-sailor I constantly mix up "lee" and "luv". Also people may understand this term from a perspective of atmospheric circulation. I think it is better to talk about "upstream" and "downstream" in these cases (and elsewhere).

We thank the reviewer for this advice. We changed the terminology for better understanding.

p6 l 6: quantify "less negative"

The sentence has been changed accordingly.

p6 l 14: This must be mentioned earlier and discussed in more detail (see general comment above).

An explanation has been added to the methods section.

p6 l 33 remove "more or less"

The sentence has been changed.

p6 l 35 when interpreting strain fields for fracture formation it is important to explicitly state the resolution of velocities and strain rates.

We thank the reviewer for this advice. The resolution of each resulting velocity field or derived quantity is 50x50m. We added this information to the respective methods part.

p7 l 16 "Even if zero is not passed, "–> "Even if values remain negative, …"

The sentence has been changed.

p7 l 29 "more or less" is not helpful here. Better write a clear sentence stating that no single stress/strain field can fully explain the observations (or something like that).

We changed the sentence for better understanding.

p7 l 30 I suggest avoiding "might be" (and "could be" later on) and instead clearly stating the uncertainties that are indicated with these formulations.

This paragraph has been re-formulated.

p7 l 36 I do not grasp the meaning of this paragraph and suggest rephrasing (see general comment above).

This paragraph has been re-structured.

p7 l 38 "certain" isoline? Is it not the 0 isoline specifying pure extension in all directions (as stated in intro)?

Doake et al. (1998) are very careful when formulating this criterion. Their figures somehow implicitly suggest that they mean the 0 isoline, but when describing the compressive arch, Doake et al. (1998) stay rather vague. There is no hard statement that the threshold is at zero. They only state that it might not deviate much from zero. Therefore we also kept their decision to not explicitly specify the zero isoline as the definition of the compressive arch. We changed the respective paragraph accordingly.

p7 l 38 "retreats/recedes" what is the difference between retreating and receding and why are both verbs mentioned here?

We opted for 'retreats' in this context.

p8 l 9 This paragraph of fractures fits to the one started in p7 l29. I also suggest to move the last paragraph of the conclusions to this section to have all fracture-related things together.

Intuitively, this comment was addressed during the re-organisation of the Results section. The passage in the conclusion was now added to the Results section where the gradual recession and evolution of WIS is recapitulated.

p 8 l 29: This paragraph brings up a observations which have not been discussed previously in the paper. I suggest not to introduce novel things in the conclusions (see previous comment).

See above answer.

p8 l 16 High variability or high uncertainty (because flow direction is ill-constrainedy?

These are uncertainties from the offset tracking that directly transmit into all derived fields. Errors are now explicitly indicated in the text.

Figure 1

I don't see the red nor the cyan line. On the other hand I see an orange ice-front position not mentioned in legend. Something went wrong here.

We are sorry for this mistake. The Figure has been changed and improved.

**References**

Burgess, E. W., Forster, R. R., Larsen, C. F. and Braun, M.: Surge dynamics on Bering Glacier, Alaska, in 2008–2011, The Cryosphere, 6(6), 1251–1262, doi:10.5194/tc-6-1251-2012, 2012.

Gray, A. L., Mattar, K. E. and Sofko, G.: Influence of ionospheric electron density fluctuations on satellite radar interferometry, Geophys. Res. Lett., 27(10), 1451–1454, doi:10.1029/2000GL000016, 2000.

Joughin, I.: Ice-sheet velocity mapping: a combined interferometric and speckle-tracking approach, Ann. Glaciol., 34(1), 195–201, 2002.

McNabb, R. W., Hock, R., O'Neel, S., Rasmussen, L. A., Ahn, Y., Braun, M., Conway, H., Herreid, S., Joughin, I., Pfeffer, W. T., Smith, B. E. and Truffer, M.: Using surface velocities to calculate ice thickness and bed topography: a case study at Columbia Glacier, Alaska, USA, J. Glaciol., 58(212), 1151–1164, doi:10.3189/2012JoG11J249, 2012.

Nagler, T., Rott, H., Hetzenecker, M., Wuite, J. and Potin, P.: The Sentinel-1 Mission: New Opportunities for Ice Sheet Observations, Remote Sens., 7(7), 9371–9389, doi:10.3390/rs70709371, 2015.

Quincey, D. J., Copland, L., Mayer, C., Bishop, M., Luckman, A. and Belo, M.: Ice velocity and climate variations for Baltoro Glacier, Pakistan, J. Glaciol., 55(194), 1061–1071, 2009.

Quincey, D. J., Braun, M., Glasser, N. F., Bishop, M. P., Hewitt, K. and Luckman, A.: Karakoram glacier surge dynamics, Geophys. Res. Lett., 38(18), L18504, doi:10.1029/2011GL049004, 2011.

Seehaus, T., Marinsek, S., Helm, V., Skvarca, P. and Braun, M.: Changes in ice dynamics, elevation and mass discharge of Dinsmoor–Bombardier–Edgeworth glacier system, Antarctic Peninsula, Earth Planet. Sci. Lett., 427, 125–135, 2015.

Williams, S., Bock, Y. and Fang, P.: Integrated satellite interferometry: Tropospheric noise, GPS estimates and implications for interferometric synthetic aperture radar products, J. Geophys. Res., 103(B11), 27051–27067, 1998.

---

## Author Comment (AC2) · 20 Jan 2017

**Point-by-point reply to review comments on 'Dynamic changes on Wilkins Ice Shelf during the 2006-2009 retreat derived from satellite observations' by Rankl et al.**

The authors want to thank both reviewers for their careful reading of the manuscript. The comments were very useful and helped to improve the manuscript substantially. Large parts of the manuscript have been re-structured following the suggestions given by Ted Scambos. More methodological aspects brought up by Reinhard Drews have been addressed extensively in the revised version of the manuscript. In the following, all comments are addressed specifically and changes in the manuscript indicated. We hope, both reviewers support the changes made in the revised manuscript.

**Comments by Ted Scambos:**

Using a time-series of InSAR and speckle-tracked radar images of velocity, Rankl et al. present very good detailed study of the events occurring on the Wilkins Ice Shelf spanning the 1990s and through the series of major calvings and disintegrations occurring in 2008 and 2009. The focus of the paper is on ice flow and strain rates as the ice shelf evolves. This analysis of events provide insight into how stresses are transferred within ice shelves, and illustrate the power of good sequential ice velocity data in diagnosing causality for riftings and calvings.

In general, the writing could be tighter. There are some odd constructions for an English first-language reader, although the meaning is clear enough. But some work on the text could probably reduce the length by 10% and make it an easier, more efficient read. In reading it, I was interested by the main data figures (Fig2, 3, and 4) but found the discussion hard to follow because of the detail – all quite accurate, but it seemed to go slowly through this part when it could have been more interesting.
We did restructure the discussion following the reviewer's suggestion below.

My main comment concerns the interpretations of Figure 2 and 3 and 4 – the data look very good, and they provide a clear story – although I see you are cautioning about data quality in slow-moving areas. What is striking is the abrupt shift in the pattern after the calving and disintegrations of Feb-March, and especially after July, 2008 – this shows that the middle section of the ice bridge was an important buttress, and that the last ice bridge whisker was already nearly detached at the northern end (see Fig 5 in Braun et al. 2009 and Fig2 in Scambos et al. 2009 – the connection to Charcot Is. is rifted and sheared prior to the removal of the central ice bridge piece). With the loss of the middle ice bridge section, strong extensional stress is present just north of the Vere ice rise, and the ice soon rifts away ..
Sections 4.1 and 4.2 work through the data in the figures slowly. . . it would be more emphatice and clear to introduce the three figures briefly (just say what they are) and then discuss the evolution of the fractures and strain rates in a more story-like fashion. Readers would retain the events and significance better.

Or perhaps open with a) brief description of the data shown in the figures, and caveats, then, b) an overview 'story' of how events proceeded and the major fractures and shifts in strain patterns, and then c) perhaps some kind of review of the details captured by the Fig3 and 4 data that you are discussing on Page 6 and 7.
We thank the reviewer for this useful suggestion. We followed the reviewer's suggestion to split the discussion into 3 parts. The changes have certainly facilitated the readability of the manuscript and highlight the main results more succinctly.

I think the data make the sudden rearrangement the highlight of the paper; they underscore the relative lack of importance of the easternmost section of the ice bridge (or, if you like, the importance of the middle section.)

Overall, I think the manuscript is nearly publishable as it is, but would benefit and be more likely to be remembered and cited with another round of editing with respect to telling the story more clearly and succinctly. The conclusions have this kind of 'voice'.
The conclusions have been restructured.

P2L1 – in several papers, I've been trying to reserve this word for the kind of fine-scale rapid calving that was observed on Larsen B in January - March 2002 and Wilkins in Feb29-March8 2008. Please use the word 'collapse' here, since the ice shelf instability caused by the loss of the compressive arch might simply result in a series of large-scale calvings spanning months or even years, and not a true 'disintegration'.

We thank the reviewer for this helpful advice. The sentence has been changed accordingly.

P3L13 SNR: This needs a bit of unpacking – what you mean is a correlation peak height that is less than 4 times the mean correlation away from the peak. 'Signal' to 'noise' is a bit obscure here.

The reviewer is right with his explanation here, however, the term signal-to-noise ratio is commonly used when measuring the confidence of the offset estimates derived from intensity offset tracking (Seehaus et al., 2015; Strozzi et al., 2002; Werner et al., 2005). Thus, we would like to keep this terminology.

P5L33-34 Yes, extensional strain, but once the rift had formed in July 2007, almost all of the 'strain' would be taken up by the rift widening. It's what the strain rate was prior to the new rift (which, agreed, would have been formed by the stress build-up).

This remark of the reviewer is very subtle as it refers somehow to the transient stress evolution during fracture opening. Certainly, the stress field evolves as rifts open. This is difficult to trace with displacement fields inferred from consecutive image pairs acquired with a time lag of many days or even months. These displacements are highly averaged, while the rearrangement might have taken place on much shorter time-scales. We added this notion to the text when introducing the results.

P7L3 – '. . .it is obvious . . .. ' this phrase is odd, the block might have just calved away intact?

This ambiguous formulation was removed during rewriting the entire section following the main reviewer comments.

P7L8-25 I find this section somewhat of a difficult read – too tentative, to qualified; there's a basic story from the data, but it's obscured by nuance here. For example, L19-21, the ice bridge has a stabilizing effect, yes, and that places it under compressive strain along its axis, thus leading to, not failing to prevent, its eventual collapse - ?

Also, this paragraph was rewritten more clearly and succinctly during re-organization of the result section.

Fig3 and Fig4 – there is a white (fig3) and red (fig4) dot near the northeastern corner of the ice bridge – what does that signify? Please describe it in the captions.

The red/white dots mark an ice rise at the northeastern corner of the ice bridge. In the revised figures we removed the dots, since this information is not relevant for the analysis.

Please also note the supplement to this comment:

We thank reviewer for his helpful comments. The manuscript has been adapted accordingly.

**References**:

Seehaus, T., Marinsek, S., Helm, V., Skvarca, P. and Braun, M.: Changes in ice dynamics, elevation and mass discharge of Dinsmoor–Bombardier–Edgeworth glacier system, Antarctic Peninsula, Earth Planet. Sci. Lett., 427, 125–135, 2015.

Strozzi, T., Luckman, A., Murray, T., Wegmuller, U. and Werner, C. .: Glacier motion estimation using SAR offset-tracking procedures, Geosci. Remote Sens. IEEE Trans. On, 40(11), 2384–2391, 2002.

Werner, C., Wegmuller, U., Strozzi, T. and Wiesmann, A.: Precision estimation of local offsets between pairs of SAR SLCs and detected SAR images, in Proceedings. 2005 IEEE International Geoscience and Remote Sensing Symposium, 2005. IGARSS'05., vol. 7, pp. 4803–4805, IEEE., 2005.

---

## Author Response (AR2)

*Again, the authors thank Reinhard Drews for the very careful reading of the manuscript and the helpful comments. Most of them have been integrated in the manuscript.*

ABSTRACT
p1 l 8: remove "can"
Corrected as suggested.
p1 l 10: replace "This present study..." with "This study.." or "Here, we.." to avoid repetition with the same opening two sentences later.
Corrected as suggested.
p1 l 12: replace "in respect" with "relativ"
Corrected as suggested.
p1 l 15: replace "the ice bridge" with "an ice bridge" (the ice bridge is has not been introduced yet)
Corrected as suggested.
p1 l 16: How about rephrasing: "We identify areas that are important for buttressing and areas prone to fracturing using in-flow and …"
Corrected as suggested.
p1 l 17: "Further propagation.." of what? (the connection to the "speed-up" mentioned two sentences earlier is lost at this stage).
Corrected as suggested.
p1 l 18: How about: "Positive second principal stresses are another scale-free indicator..."
Corrected as suggested.
p1 l 19: remove "rather"
Corrected as suggested.
Comment: I am somehow missing a punchy last sentence in the Abstract about the main conclusion/implications of this study.
A concluding remark has been added to the abstract as well as the conclusions.

INTRODUCTION
p1 l 30: move reference Paolo et al., 2015 to l 29 (after "thinning rates") to avoid the double-brackets.
Corrected as suggested.
p1 l 35: I suggest to remove "As it is long known..." and start right away with "There were regular attempts..".
Corrected as suggested.
p2 l 2: "The smaller, second principal.."
Corrected as suggested.
p2 l 7: "the the"
Corrected as suggested.
p2 l 8: replace "not very stable" with "unstable"
Corrected as suggested.
p2 l 10: replace "stability issue" with "ice-sheet stability" (if this is what you mean)
We replaced issue with criteria
p2 l 11: "shelf-ice area" or "ice-shelf area" ? I think it should be the latter.
Corrected as suggested.
p2 l 12: Remove sentence: "For the Bellinghausen Sea...." this is not relevant here.
Corrected as suggested.
p2 l 13: replace second "during" with "over" to avoid repetition
Corrected as suggested.
p2 l 16: "..first assessment of previously invoked stability criteria."
Corrected as suggested.

STUDY AREA
p2 l18: Consider a better section heading. Maybe something like "Regional setting and history of the WIS"
Corrected as suggested.
p2 l35: Remove "Continous" with "The"
Corrected as suggested.
p2 l35: Replace "ice connection" with "ice bridge" in all instances to use coherent terminology with Figure 1 and the later text/abstract.

Corrected as suggested.

p3 l7: What is the relevance of the released energy in general, and for this paper in particular? I suggest removal of this sentence.

Corrected as suggested.

SURFACE VELOCITIES

p3 l 21: Replace "output spatial resolution" with "spatial gridding". I also don't fully agree that the strain rates are resolved on a 50 m grid, because you apply heavy smoothing (1x1 km) to avoid noise amplification. Saying they were "gridded" to 50 m is more correct.

Corrected as suggested.

p3 l 25: Ice on ice rises is not "non-moving". It is maybe "slow-moving" (near divides with several meters per year).

Further specification has been added.

p3 l 34: Remove "Of course,"

Corrected as suggested.

p3 l 37: Replace "most important" with "most significant" implying that the flow differences are larger than the observational errors.

Corrected as suggested.

p4 l 20: Replace "also true" with "comparable with"

Corrected as suggested.

STRESS TENSOR AND STRAIN RATES

p4 l 29 Shorten: "Ice is typically described as a non-Newtonian.." (there are many possibilities like this to make the paper more succinct).

However, we consider this as a useful information in this context.

p4 l 31: "The constitutive equation.." (remove "In its most general form")

Corrected as suggested.

p5 l 2: replace "in case there is no" with "in areas without"

Corrected as suggested.

p5 l 5: is it not "subtracting"?

The sentence has been changed.

p5 l 7: "where satellite data were acquired" (data is plural)

Corrected as suggested.

p5 l 16: WATCH OUT: I understand that second invariant of a matrix A is $0.5*(tr(A)^2-tr(A^2))$ so there should be some off-diagnal elements in there. I don't see this in your revised description which uses only "ii" indices. Did I miss something?

The equation has been changed accordingly.

p5 l 29: "the change in them" with "temporal changes"

Corrected as suggested.

OVERVIEW

p6 l 7: Consider more informative section heading.

The heading has been changed.

p6 l 9: The ice shelf is mainly fed by surface accumulation (as stated in abstract)

The sentience as restructured.

p6 l 10: replace "once the grounding line is passed" with "seawards of the grounding line"

Corrected as suggested.

p6 l 11: "quickly" is temporal. Maybe "abruptly"?.

Corrected as suggested.

p6 l 26 "mostly limit the discussion" ? Remove "mostly". Maybe "We limit the discussion to...unless specified otherwise".

Corrected as suggested.

p7 l 1: "This stress-flow angle was introduced..." and the following sentence are repetitive to intro. Consider streamlining.

This section was shortened as suggested.

p7 l 8: Replace "In this present study.." with "Here, we…"

Corrected as suggested.

RETREAT STAGES

p7 l 13: Is the 1993 break-up mentioned in Section 2? If not, it should be mentioned there.

The ice front position in 1993 is shown in Figure 1 and the text refers to it.

p8 l 5: "which was the reason for why" remove "for"
Corrected as suggested.

A PRORI INDICATIONS...
p9 l 7" "we want to follow" replace "we follow"
Corrected as suggested.
p9 l 16: this paragraph is now much clearer.

CONCLUSIONS
p9 l 35 "rather insensitive" "rather unfavorable" there is a lot of rathers throughout the manuscript.
Corrected as suggested.
Same as for the abstract I suggest to end the conclusions with a more positive note summarizing the main conclusions/implications.
The last paragraph of the conclusions has been changed following the suggestion above.

Figure S2: Shouldn't the vertical displacement of point F be zero (i.e. is the profile A-A' flipped left/right)?
The figure has been changes accordingly. We apologize for this mistake.

[revised manuscript text omitted]